# Robust Thompson Sampling for Gaussian Bandits against Reward Poisoning Attacks

## Abstract

Thompson sampling is one of the most popular learning algorithms for online sequential decision-making problems and has rich real-world applications. However, current Thompson sampling algorithms are limited by the assumption that the rewards received are uncorrupted, which may not be true in real-world applications where adversarial reward poisoning exists. To make Thompson sampling more reliable, we want to make it robust against adversarial reward poisoning. The main challenge is that one can no longer compute the actual posteriors for the true reward, as the agent can only observe the rewards after corruption. In this work, we solve this problem by computing pseudo-posteriors that are less likely to be manipulated by the attack. Particularly, we focus on two popular settings: stochastic bandits and contextual linear bandits with priors as Gaussian distributions. We propose robust algorithms based on Thompson sampling for the popular stochastic and contextual linear bandit settings in both cases where the agent is aware or unaware of the budget of the attacker. We theoretically show that our algorithms guarantee near-optimal regret under any attack strategy.

## 1 Introduction

The multi-armed bandit (MAB) setting is a popular learning paradigm for solving sequential decision-making problems (Slivkins et al., 2019). The stochastic and linear contextual MAB settings are the most fundamental and representative of the different bandit settings. Due to their simplicity, many industrial applications such as recommendation systems frame their problems as stochastic or contextual linear MAB (Brodén et al., 2018; Chu et al., 2011). As one of the most famous stochastic bandit algorithms, Thompson sampling has been widely applied in these applications and achieves excellent performance both empirically (Chapelle & Li, 2011; Scott, 2010) and theoretically (Agrawal & Goyal, 2013; 2017). Compared to another popular exploration strategy known as optimality in the face of uncertainty (OFUL/UCB), Thompson sampling has several advantages:

- **Utilizing prior information**: By design, Thompson sampling algorithms utilize and benefit from the prior information about the arms.

- **Easy to implement**: While the regret of a UCB algorithm depends critically on the specific choice of upper-confidence bound, Thompson sampling depends only on the best possible choice. This becomes an advantage when there are complicated dependencies among actions, as designing and computing with appropriate upper confidence bounds present significant challenges Russo et al. (2018). In practice, Thompson sampling is usually easier to implement Chapelle & Li (2011).

- **Stochastic exploration**: Thompson sampling is a random exploration strategy, which could be more resilient under some bandit settings Lancewicki et al. (2021).

Despite the success, Thompson sampling faces the problem of low efficacy under adversarial reward poisoning attacks Jun et al. (2018); Xu et al. (2021); Liu & Shroff (2019). Existing algorithms assume that the reward signals corresponding to selecting an arm are drawn stochastically from a fixed distribution depending on the arm. However, this assumption does not always hold in the real world. For example, a malicious user can provide an adversarial signal for an article from a recommendation system. Even under small corruption, Thompson sampling algorithms suffer from significant regret under attacks. While robust versions of the learning algorithms following

other fundamental exploration strategies such as optimality in the face of uncertainty (OFUL) and $\epsilon$-greedy were developed Lykouris et al. (2018); Neu & Olkhovskaya (2020); Ding et al. (2022); He et al. (2022); Xu et al. (2023), there has been no prior investigation of robust Thompson sampling algorithms. The main challenge is that under the reward poisoning attacks, it becomes impossible to compute the actual posteriors based on the true reward, which is essentially required by the algorithm. Naively computing the posteriors based on the corrupted reward causes the algorithm to be manipulated by the attacker arbitrarily (Xu et al., 2021).

**This work**. We are the first to show the feasibility of making Thompson sampling algorithms robust against adversarial reward poisoning. We focus on two popular bandit settings: stochastic bandits and contextual linear Gaussian bandits, and the prior distributions of each arm are Gaussian distributions. Our main contribution is developing robust Thompson sampling algorithms for the two bandit settings. We consider both cases where the corruption budget of the attack is known or unknown to the learning agent. The regrets induced by our algorithms under the attack are near-optimal with theoretical guarantees. We adopt two ideas to achieve robustness against reward poisoning attacks in the two MAB settings. The first idea is 'optimality in the face of corruption,' a general idea similar to the popular exploration strategy, 'optimality in the face of uncertainty.' In "optimality in the face of uncertainty," the agent optimistically estimates the best possible reward for an arm considering the uncertainty in its evaluation. Similarly, in "optimality in the face of corruption", the agent optimistically estimates the best possible reward for an arm considering the uncertainty and the influence of data corruption on its estimation. In the stochastic MAB setting, we show that the Thompson sampling algorithm can maintain sufficient explorations on arms and identify the optimal arm by relying on optimistic posteriors considering potential attacks. The second idea is to adopt a weighted estimator He et al. (2023) that is less susceptible to the attack. In the linear contextual MAB setting, we show that with such an estimator, the influence of the attack on the estimation of the posteriors is limited, and the Thompson sampling algorithm can almost always identify the optimal arm at each round with a high probability. We empirically demonstrate the training process of our algorithms under the attacks and show that our algorithms are much more robust than other fundamental bandit algorithms, such as UCB, in practice. Compared to the state-of-the-art robust algorithm CW-OFUL He et al. (2022) for linear contextual bandit setting, our algorithm is as efficient, and in addition, it inherits the advantages from using Thompson sampling exploration strategy as aforementioned.

## 2 RELATED WORK

**Thompson Sampling Algorithms:** Algorithms based on the Thompson sampling exploration strategy have been widely applied in online sequential decision-making problems Agrawal et al. (2017); Bouneffouf et al. (2014); Ouyang et al. (2017), including MAB settings with different constraints and reinforcement learning. Agrawal & Goyal (2013; 2017) develop insightful theoretical understandings of the learning efficiency of Thompson sampling. In the stochastic MAB and linear contextual MAB settings, algorithms based on Thompson sampling achieve near-optimal performance in that the upper bounds on regret match the lower bound of the setting Agrawal & Goyal (2012; 2013). However, these algorithms are designed for a setting without poisoning attacks, which may not be true in real-world scenarios.

**Reward Poisoning Attack against Bandits:** Reward poisoning attacks against bandits have been considered a practical threat against MAB algorithms Lykouris et al. (2018). A majority of studies on poisoning attacks adopt a strong attack model where the attacker decides its attack strategy after the agent takes an arm at each timestep Jun et al. (2018); Liu & Shroff (2019); Garcelon et al. (2020). This attack scenario is argued to be more practical Zhang et al. (2021). Jun et al. (2018) proposes attack strategies that can work for specific learning algorithms. It has been well understood that the most famous bandit algorithms are vulnerable to poisoning attacks. The other attack model is called weak attack, where the attacker decides its attack strategy before the agent takes an arm Lykouris et al. (2018). Xu et al. (2021) shows that a family of algorithms, including the most famous ones, are vulnerable even under weak attacks.

**Robust Bandit Algorithms:** Finding bandit algorithms robust against poisoning attacks is a popular topic. Robust algorithms against weak or strong attack models have been developed in both stochastic bandit and linear contextual bandit settings Lykouris et al. (2018); Neu & Olkhovskaya (2020); Ding

et al. (2022); He et al. (2022). Recently, Wei et al. (2022) shows that with an algorithm robust against the strong attack model, one can extend it to a robust algorithm against the weak attack model. Our work focuses on robustness against the strong attack model as (1) the strong attack model is more practical, and (2) one can develop robust algorithms against weak attacks based on our algorithms. We compare the theoretical guarantees of our robust algorithms and current state-of-the-art robust algorithms in Table 5.

**Differentially Private Bandits:** Differentially-private bandit setting is closely related to the poisoning attack (Mishra & Thakurta, 2015; Hu et al., 2021), and efficient learning algorithms have been achieved through Thompson sampling (Hu & Hegde, 2022). The differentially private setting can be considered a different robustness against poisoning attack setting. Here, the attacker can modify a certain number of data points, and a differentially private algorithm must ensure its behavior is similar under any possible attacks. However, there are two main differences between the two settings: 1. The constraint on the attack. In our reward poisoning setting, the attacker can poison as much data as wanted, as long as the total amount of perturbation is limited. In contrast, in the differentially private setting, the attacker can only poison a limited number of data points. 2. The goal of the learning algorithm is different. In the reward poisoning setting, the agent only needs to guarantee that the total regret is limited under the corruption. In contrast, in the differentially private setting, the agent should behave almost identically when some data are corrupted. As a result, a differentially private algorithm is not necessarily a robust algorithm against reward poisoning attacks. Even under a limited corruption budget, the observed data can completely differ from the original data at every data point during training. So, the guarantees on differential privacy cannot directly lead to guarantees on a tight bound of regret under the reward poisoning attack.

## 3 PRELIMINARIES

### 3.1 STOCHASTIC BANDIT SETTING

For the stochastic multi-armed bandit setting, an environment consists of $N$ arms with fixed support in $[0, 1]$ reward distributions centered at $\{\mu_1, \ldots, \mu_N\}$, and an agent interacts with the environment for $T$ rounds. At each round $t$, the agent selects an arm $i(t) \in [N]$ and receives reward $r_t^o$ drawn from the reward distribution associated with the arm $i(t)$. The performance of the bandit algorithm is measured by its expected regret $R_T = \mathbb{E}\left[\sum_{t=1}^{T} \left(\mu_{i^*} - \mu_{i(t)}\right)\right]$, where $i^*$ is the best arm at hindsight, i.e., $i^* = \arg\max_{i \in [N]} \mu_i$. Without loss of generality, we assume arm $i^* = 1$ is the optimal arm. We denote $\Delta_i = \mu_1 - \mu_i$ as the gap between arm $i$ and the optimal arm. The time horizon $T$ is predetermined, but the reward distribution of each arm is unknown to the agent. The agent's goal is to minimize its expected regret $R_T$.

### 3.2 LINEAR CONTEXTUAL BANDIT SETTING

Next, we consider the linear contextual bandit setting. An environment consists of $N$ arms and a context space with $d$ dimensions $\mathbb{R}^d$, and an agent interacts with the environment for $T$ rounds. At each round $t$, $N$ contexts $\{x_i(t) \in \mathbb{R}^d\}, i = 1, \ldots, N$ are revealed by the environment for the $N$ arms. We denote $\mathbf{x}(t) = (x_1(t), \ldots, x_N(t))$. The agent draws an arm $i(t)$ and receives a reward $r_i^o(t)$. The reward is drawn from a distribution dependent on the arm $i(t)$ and the context $x_{i(t)}(t)$. In the linear contextual bandit setting, the expectation of reward is a linear function depending on the context: $\mathbb{E}[r(t)|x_{i(t)}(t)] = x_{i(t)}(t)^T \mu$, where $\mu \in \mathbb{R}^d$ is the reward parameter. Without loss of generality, we assume that the contexts and the reward parameters are bounded $\|x_i(t)\|_2 \le 1, \|\mu\|_2 \le 1$. The regret to measure the performance of the agent in this case is defined as $R_T = \sum_{t=1}^{T} x_{i^*(t)}(t)^T \mu - x_{i(t)}(t)^T \mu$ where $i^*(t) = \arg\max_i x_i(t)^T \mu$ is the optimal arm at time $t$. The time horizon $T$ is predetermined, but the agent's reward parameter $\mu$ is unknown. The goal of the agent is to minimize the regret.

To make the regret bounds scale-free, we adopt a standard assumption Agrawal & Goyal (2013) that $\epsilon_t = r^o(t) - x_{i(t)}(t)^T \mu$ is conditionally $\sigma$-sub-Gaussian for a constant $\sigma \ge 0$, i.e.,

$$\forall \lambda \in \mathbb{R}, \mathbb{E}\left[e^{\lambda \epsilon_t} \mid \{x_i(t)\}_{i=1}^{N}, \mathcal{H}_{t-1}\right] \le \exp\left(\frac{\lambda^2 \sigma^2}{2}\right),$$

where $\mathcal{H}_{t-1} = \{i(s), r(s), x_{i(s)}(s), s = 1, \ldots, t-1\}$.

### 3.3 STRONG REWARD POISONING ATTACK AGAINST BANDITS

This work considers the strong reward poisoning attack model Wei et al. (2022), where an attacker sits between the environment and the agent. At each round $t$, the attacker observes the arm pulled by the agent $i(t)$, the reward $r^o(t)$, and additionally the context $x_{i(t)}(t)$ in the contextual bandit setting. Then the attacker can inject a perturbation $c(t)$ to the reward, and the agent will receive the corrupted reward $r(t) = r^o(t) + c(t)$ in the end. The attacker has full knowledge of the environment and the learner, including the algorithm it uses and the actions it takes each time. We denote $C$ as the budget of the attack $C : \sum_{t=1}^{T} |c(t)| \leq C$, representing the maximum amount of total perturbation it can apply during the training process. We also refer to it as 'corruption level' since it indicates the level of corruption the learning agent faces.

The weak attack model has been considered in previous works Lykouris et al. (2018); Liu & Shroff (2019). Unlike the strong attack model, the weak attack has to decide on the perturbation of the reward before observing the actions taken by the agent. In this work, we only consider the strong attack model for the following reasons: 1. in practice, the strong attack is more realistic. For example, in a recommendation system, the attacker, which is a malicious user, observes the recommendation first before deciding on the malicious feedback 2. Wei et al. (2022) shows that a robust algorithm against strong attack can be used to construct robust algorithms against weak attacks. In section 4 and 5, we discuss the case where the corruption level or an upper bound on it is known or unknown to the learning agent.

### 3.4 THOMPSON SAMPLING ALGORITHMS

Thompson sampling is a heuristic exploration strategy that belongs to the family of randomized probability matching algorithms Thompson (1933). At each time step, a general Thompson sampling algorithm takes an arm based on a randomly drawn belief about the rewards of the arms. More specifically, the algorithm maintains a posterior distribution related to the expected reward of each arm. At each time, the algorithm samples a parameter from each arm's posterior to formulate a belief on the reward of an arm, and then it takes the arm with the maximal reward belief. After observing the reward of the taken arm, the algorithm updates the posteriors. The distribution of each arm's prior will influence the exact format of the algorithm.

In this work, we always assume that the priors for the rewards of the arms and the reward parameter are Gaussian distributions. This is a typical choice representative of Thompson sampling algorithms Agrawal & Goyal (2013; 2017). Although our algorithms assume Gaussian priors, in principle, it is not hard to extend them to other kinds of priors following the same idea, and many of our theoretical results are not dependent on the format of priors. In the Appendix, we show the Thompson sampling algorithms in Alg 3 and 4 for the stochastic and linear contextual MAB settings, respectively, with Gaussian distributions as priors.

## 4 ROBUST THOMPSON SAMPLING FOR STOCHASTIC MULTI-ARMED BANDITS

In this section, we present our robust Thompson sampling algorithm for the stochastic MAB setting and the theoretical guarantee of its learning efficiency. We discuss both cases, whether the corruption level $C$ is known or unknown to the learning agent. To understand why the original Thompson sampling algorithm (Alg 3) is vulnerable under the poisoning attack, we note that the actual posteriors of arms given the uncorrupted reward drawn from the environments can no longer be acquired. When calculating the posterior as if the data are uncorrupted, the resulting posteriors can be biased to the actual posteriors. Therefore, the attacker can make the learner underestimate the posteriors of the optimal arms or overestimate that of the sub-optimal arms. As a result, the learner believes that the optimal arm has a low reward and rarely selects it.

To make the algorithm robust against attack, we utilize the idea of optimism. Instead of computing the posteriors as if the data are uncorrupted, the algorithm computes the optimistic posteriors with respect to corruption for each of the arms. For each arm, the algorithm finds the posterior that maximizes the expected reward for any possible true rewards before corruption. In the stochastic MAB settings with

Gaussian priors, the mean in the posterior is $\frac{\sum_{s=1,i(s)=i}^{t-1} r(s)}{k_i(t)+1}$ where $k_i(t)$ is the number of times arm $i$ be pulled before time $t$. Since the possible reward with the maximal sum is $\sum_{s=1,i(s)=i}^{t-1} r(s) + C$, the mean of the optimistic posterior is $\frac{\sum_{s=1,i(s)=i}^{t-1} r(s)+C}{k_i(t)+1}$.

The robustness against the bias of posteriors induced by the poisoning attack is achieved by optimism. By using the optimistic posteriors for each arm, the algorithm never underestimates the posteriors of any arm. Even if a sub-optimal arm becomes the empirically optimal arm, after being selected a few times, its optimal posterior will be close to its actual posterior, which is inferior to the optimistic posterior of the optimal arm. As a result, the optimal arm will eventually be selected. Together with the Thompson sampling strategy to deal with the stochastic rewards from the environment, the algorithm can be robust and efficient in the stochastic MAB setting under poisoning attacks.

The empirical post-attack mean $\hat{\mu}_i(t)$ for arm $i$ at time $t$ is defined by $\hat{\mu}_i(t) := \frac{\sum_{s=1,i(s)=i}^{t-1} r(s)}{k_i(t)+1}$ (note that $\hat{\mu}_i(t) = 0$ when $k_i(t) = 0$) and the empirical pre-attack mean $\hat{\mu}_i^o(t)$ for arm $i$ at time $t$ is defined by $\hat{\mu}_i^o(t) := \frac{\sum_{s=1,i(s)=i}^{t-1} r^o(s)}{k_i(t)+1}$ (note that $\hat{\mu}_i^o(t) = 0$ when $k_i(t) = 0$). Let $\theta_i(t)$ denote a sample generated independently for each arm $i$ from the posterior distribution at time $t$. This is generated from posterior distribution $\mathcal{N}\left(\hat{\mu}_i(t) + \frac{\overline{C}}{k_i+1}, \frac{1}{k_i(t)+1}\right)$, where $\overline{C}$ is a hyper-parameter of the algorithm for tuning robustness against different level of corruption. In Alg 5, we formally show our robust Thompson sampling algorithm for the stochastic MAB setting.

---

**Algorithm 1** Robust Thompson Sampling for Stochastic Bandits

1: **Params**: robustness level $\overline{C}$
2: For all $i \in [N]$, set $k_i = 0, \hat{\mu}_i = 0$
3: **for** $t = 1, 2, \ldots,$ **do**
4:     For each arm $i = 1, \ldots, N$, sample $\theta_i(t)$ from the $\mathcal{N}\left(\hat{\mu}_i + \frac{\overline{C}}{k_i(t)+1}, \frac{1}{k_i(t)+1}\right)$ distribution.
5:     Play arm $i(t) := \arg\max_i\{\theta_i(t)\}$ and observe reward $r_t$
6:     Set $\hat{\mu}_{i(t)} := \frac{\hat{\mu}_{i(t)} k_i(t)+r_t}{k_{i(t)}+1}, k_{i(t)} := k_{i(t)} + 1$
7: **end for**

---

At each round $t$, Alg 5 samples a parameter $\theta_i(t)$ from a Gaussian distribution for arm $i$ with the compensation term $\frac{\overline{C}}{k_i(t)+1}$ to make it the optimistic posterior. This enables the algorithm to explore the optimal arm even if the attack injects a negative bias. Notice that when $\overline{C} = 0$, it degenerates into original Thompson Sampling using Gaussian priors. The regret of the algorithm under the attack is guaranteed in Theorem 4.1 as below.

**Theorem 4.1.** *For the $N$-armed stochastic bandit problem under any reward poisoning attack with corruption level $C$, the expected regret of the Robust Thompson Sampling Alg 5 with $\overline{C} \geq C$ is bounded by:*

$$\mathbb{E}[\mathcal{R}(T)] \leq O(\sqrt{NT \ln N} + N\overline{C} + N)$$

*The big-Oh notation hides only absolute constants.*

**Proof Sketch:** A detailed proof can be found in the Appendix. Our proof technique is based on a previous study (Agrawal & Goyal, 2017). Here, we provide a sketch for the proof. First, we define two good events:

**Definition 4.2** (Good Events). *For $i \neq 1$, define $E_i^\mu(t)$ is the event $\hat{\mu}_i(t) \leq \mu_i + \frac{\Delta_i}{3}$, and $E_i^\theta(t)$ is the event $\theta_i(t) \leq \mu_i - \frac{\Delta_i}{3}$.*

$E_i^\mu(t)$ represents the case where the empirical means of sub-optimal arms are not much larger than their true mean. $E_i^\theta(t)$ represents cases where the sampled rewards from sub-optimal arms' posteriors are less than their true mean. Next, we can prove that when both good events are true, the probability that the agent selects the optimal arm is high, so the regret in this case is limited. Then, we can prove that the probability of either or both good events being false is low, so even if the worst scenario happens when the good events are false, the regret is still limited due to the low probability of this

case. The key reason why both good events are true with a high probability is because of the bonus term $\frac{\overline{C}}{k_i(t)s+1}$ we add for the posterior distributions of each arm compensates for the bias induced by the attack in the worst case. As a result, the agent is very unlikely to underestimate the performance of each arm under any poisoning attack within the budget limit. Therefore, the explorations of each arm are likely to be sufficient. Finally, by combining all the cases, we can show that the total regret is limited.

**Corruption level $C$ known to the learner:** In this case, we simply set $\overline{C} = C$ in Alg 5. Then, the dependency of regret on $C$ is linear according to Theorem 4.1, which is near-optimal Gupta et al. (2019).

**Corruption level $C$ unknown to the learner:** In this case, we set $\overline{C} = \sqrt{T \ln N / N}$ in Alg 5. Theorem 4.1 shows that if $C \leq \sqrt{T \ln N / N}$, the regret can be upper bounded by $O(\sqrt{NT \ln N})$ when $T \geq N$, and if $C > \sqrt{T \ln N / N}$ the regret can be trivially bounded by $O(T)$. This upper bound is near-optimal when $C \leq \sqrt{T \ln N / N}$ due to the lower bound in Theorem 1.4 from Agrawal & Goyal (2017). And the multi-armed bandit case of Theorem 4.12 in He et al. (2023) shows it's also near-optimal when $C > \sqrt{T \ln N / N}$.

## 5 ROBUST THOMPSON SAMPLING FOR CONTEXTUAL LINEAR BANDITS

In this section, we present our robust Thompson sampling algorithm for the contextual linear MAB setting and the theoretical guarantee of its learning efficiency. The vulnerability of Thompson sampling in the linear contextual bandit setting is similar to that in the stochastic bandit setting. The posterior on the reward parameter based on the corrupted reward can be biased compared to the actual posterior, resulting in poor decisions on action selection. Even worse, the bias induced by the reward corruption is relatively large when computing the posterior parameters as in Alg 4. Consequently, Zhao et al. (2021) follows the UCB exploration strategy using such an estimator, and the resulting algorithm is still not robust enough under the poisoning attacks. Therefore, we are not using the original estimator for our robust algorithm.

Inspired by He et al. (2023), we use a weighted ridge regression estimator as described in Alg 2 line 6 to compute the expectation of the Gaussian posterior. Such an estimator can effectively reduce the bias induced by reward poisoning. The key is that it assigns less weight $w_t$ to the data with a 'large' context $w_t = \min\left\{1, \gamma / \left\|x_{i(t)}(t)\right\|_{B(t)^{-1}}\right\}$ so that its estimation is less sensitive to data corruption in these cases. We formally show the algorithm in Alg 2, where $v_t = \sigma \sqrt{9d \ln\left(\frac{t+1}{\delta}\right)}$ and $\gamma > 0$ is a hyper-parameter representing the degree of robustness against different corruption.

---

**Algorithm 2** Robust Thompson Sampling for Contextual Linear Bandits

1: **Params**: robustness level $\gamma$
2: Set $B = I_d, \hat{\mu} = 0_d, f = 0_d$.
3: **for** $t = 1, 2, \ldots,$ **do**
4:     Sample $\tilde{\mu}(t)$ from distribution $\mathcal{N}\left(\hat{\mu}, v_t^2 B(t)^{-1}\right)$.
5:     Play arm $i(t) := \arg\max_i x_i(t)^T \tilde{\mu}(t)$, and observe reward $r_t$.
6:     Set $w_t = \min\left\{1, \gamma / \left\|x_{i(t)}(t)\right\|_{B(t)^{-1}}\right\}$
7:     Update $B(t+1) = B(t) + w_t x_{i(t)}(t) x_{i(t)}(t)^T, f = f + w_t x_{i(t)}(t) r_t, \hat{\mu} = B(t)^{-1} f$.
8: **end for**

---

In general, Alg 2 is very similar to the original version in Alg 4 except for using the ridge regression estimator. This change ensures that the posterior calculated in line 3 is not far from the actual posterior under poisoning attacks. In Theorem 5.1, we provide a high probability bound on the regret for Alg 2.

**Theorem 5.1.** *For the stochastic contextual bandit problem with linear payoff functions, with probability $1 - \delta$, the total regret in time $T$ for Robust Thompson Sampling for Contextual Linear*

*Bandits (Algorithm 2) is bounded by*

$$O\left(de^{(1+\frac{C\gamma}{\sqrt{d}})^2}\sqrt{dT\ln T\ln\left(\frac{T}{\delta}\right)} + C\gamma e^{(1+\frac{C\gamma}{\sqrt{d}})^2}\sqrt{dT\ln T}\right.$$

$$\left. + \frac{d^2 e^{(1+\frac{C\gamma}{\sqrt{d}})^2}}{\gamma}\ln T\sqrt{\ln\left(\frac{T}{\delta}\right)} + Cde^{(1+\frac{C\gamma}{\sqrt{d}})^2}\ln T\right)$$

**Proof Sketch:** Here we provide a proof sketch for Theorem 5.1. The full proof can be found in the Appendix. Similar to the proof for Theorem 4.1, first, we define two good events:

**Definition 5.2** (Good Events). Define $E^\mu(t)$ as the event

$$\forall i : \left|x_i(t)^T\hat{\mu}(t) - x_i(t)^T\mu\right| \leq \left(\sigma\sqrt{d\ln\left(\frac{t^3}{\delta}\right)} + 1 + C\gamma\right)\|x_i(t)\|_{B(t)^{-1}}$$

Define $E^\theta(t)$ as the event that

$$\forall i : \left|\theta_i(t) - x_i(t)^T\hat{\mu}(t)\right| \leq \sqrt{4d\ln(t)}v_t\|x_i(t)\|_{B(t)^{-1}}.$$

$E^\mu(t)$ represents the case where the mean of the posterior of each arm is close to its actual reward parameter. $E^\theta(t)$ represents the case where the sampled reward for each arm is close to its expected value. Next, we prove that both good events hold with a high probability. The key reason is that the posterior distribution computed by the weighted estimator is less sensitive to any change in the rewards. Therefore, the agent is more robust against data corruption. Therefore, the evaluation for each arm is more likely to be accurate, and the exploration will be sufficient correspondingly. For each arm that is sub-optimal at a round, we define an arm as saturated if it has been selected more than a specific number of times. The value for this particular number is also limited due to the weighted estimator. Next, we prove that if an arm is saturated and both good events are true, then it is very unlikely for the algorithm to select the arm. In other words, for a sub-optimal arm at a round, if it has already been selected a limited number of times, then it is very unlikely to be chosen furthermore unless the good events are false, which happens with a low probability. Therefore, the cumulative times that a sub-optimal arm is selected at a time is limited, so the regret of the algorithm is bounded.

**Corruption level $C$ known to the learner:** In this case, we set $\gamma = \sqrt{d}/C$. By Theorem 5.1, the regret is upper bounded by

$$\mathcal{R}(T) = O\left(d\sqrt{dT\ln T\ln\left(\frac{T}{\delta}\right)} + Cd\sqrt{d}\ln T\sqrt{\ln\left(\frac{T}{\delta}\right)}\right).$$

$$= \widetilde{O}(d\sqrt{dT} + d\sqrt{d}C)$$

The dependency of regret on the corruption level $C$ is near-optimal Bogunovic et al. (2021), and the algorithm becomes the same as the LinUCB algorithm when there is no corruption $C = 0$.

**Corruption level $C$ unknown to the learner:** In this case, we set $\gamma = \sqrt{d}/\sqrt{T}$ in Alg 2. From Theorem 5.1 and the results in the known $C$ case above, the algorithm's regret is upper bounded by $\widetilde{O}(d\sqrt{dT})$ when $C \leq \sqrt{T}$, else it is trivially bounded by $O(\sqrt{T})$. According to Hamidi & Bayati (2020), the worst-case lower bound for Thompson Sampling is $\Omega(d\sqrt{dT})$. Therefore, our upper bound is near-optimal when $C \leq \sqrt{T}$, and from Theorem 4.12 in He et al. (2022) we know that it's also optimal when $C > \sqrt{T}$.

In Table 5, we compare the theoretical guarantees between our robust Thompson sampling (RTS) algorithms and other state-of-the-art robust algorithms in different bandit settings.

## 6 SIMULATION RESULTS

In this section, we show the simulation results of running our algorithm on general bandit environments under typical attacks commonly used in other literature. We notice that the constant term in

| Bandit Setting | Algorithm | C | Adversary | Regret |
|---|---|---|---|---|
| Stochastic | RTS | Known | Strong | $\tilde{O}(\sqrt{NT} + NC)$ |
| Stochastic | RTS | Unknown | Strong | $\tilde{O}(\sqrt{NT}), C \leq \sqrt{T \ln N / N}$ $\tilde{O}(T), C > \sqrt{T \ln N / N}$ |
| Stochastic | MAAER | Unknown | Weak | $\tilde{O}(\sqrt{NT} + N\sqrt{CT})$ |
| Stochastic | BARBAR | Unknown | Weak | $\tilde{O}(\sqrt{NT} + NC)$ |
| Contextual | RTS | Known | Strong | $\tilde{O}(d\sqrt{dT} + d\sqrt{d}C)$ |
| Contextual | CW-OFUL | Known | Strong | $\tilde{O}(d\sqrt{T} + dC)$ |
| Contextual | RTS | Unknown | Strong | $\widetilde{O}(d\sqrt{dT}), C \leq \sqrt{T}$ $\widetilde{O}(T), C > \sqrt{T}$ |
| Contextual | CW-OFUL | Unknown | Strong | $\widetilde{O}(d\sqrt{T}), C \leq \sqrt{T}$ $\tilde{O}(T), C > \sqrt{T}$ |

Table 1: Comparison between different robust bandit algorithms. RTS is our robust Thompson sampling algorithm; MAAER is the multi-layer active arm elimination race algorithm from Lykouris et al. (2018); BARBAR is the robust algorithm from Gupta et al. (2019). CW-OFUL is the robust algorithms from He et al. (2022).

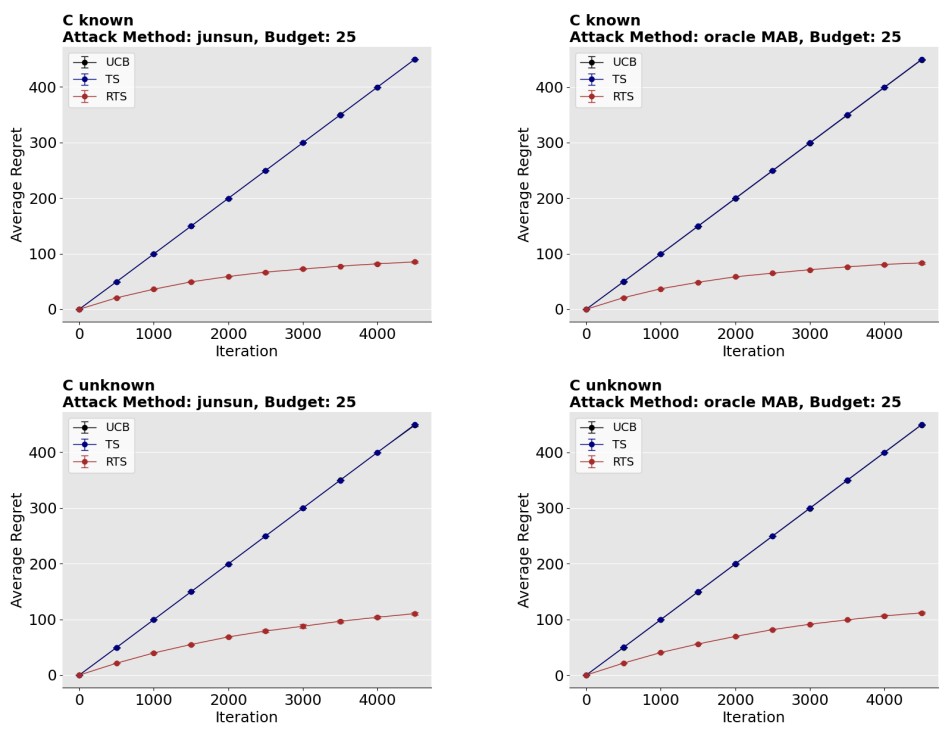

Figure 1: In the stochastic bandit setting, the cumulative regret of different learning algorithms during training under different attacks.

our regret analysis is large, so we want to show that the constant term is low in practice and verify that the regret is indeed linearly dependent on the corruption budget. We also want to use empirical results to intuitively show how our robust algorithms perform under the poisoning attack.

## 6.1 STOCHASTIC MAB SETTING

**Experiment setup:** We consider a stochastic MAB environment consisting of $N = 5$ arms and a total number of timesteps of $T = 5000$. We adopt two attack strategies: (1) Junsun's attack proposed by Jun et al. (2018). (2) oracle MAB attack, which is also used in Jun et al. (2018). The choice for

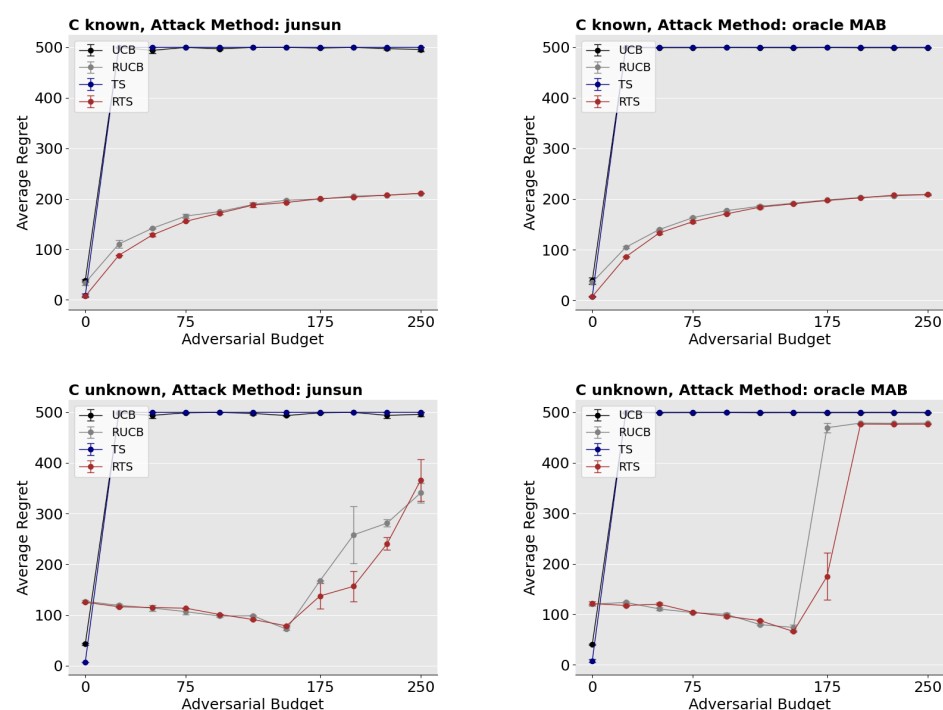

Figure 2: In the stochastic bandit setting, the total regret of different algorithms under attacks at different corruption level.

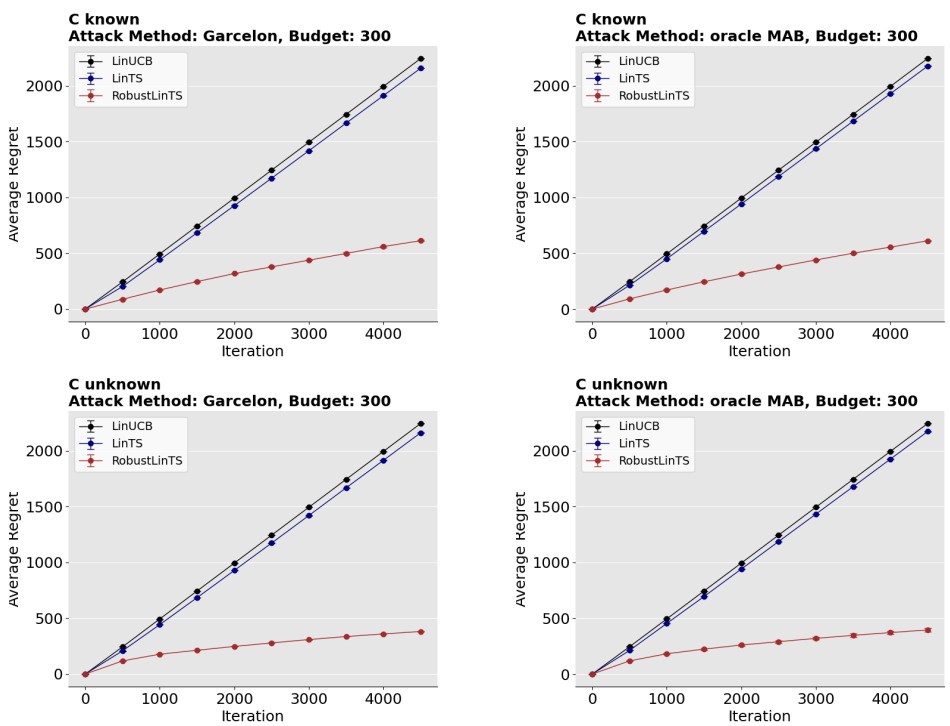

Figure 3: In the linear contextual bandit setting, the cumulative regret of different learning algorithms during training under different attacks.

corruption level varies in our experiments, which will be specified in the experimental results. Each experiment is repeated for 10 times, and we show the empirical mean and standard deviation in the experimental results.

**Cumulative regret during training under attack:** Here, we intuitively show the behavior of our robust Thompson sampling algorithms under the attack. For comparison, we choose the UCB and Thompson sampling algorithms to represent the behavior of an efficient but not robust algorithm. The corruption level for the attack is set as $C = 25$. Such a corruption level is high enough to show the vulnerability of a vulnerable attack and lower than the threshold to induce high regret on a robust algorithm when $C$ is unknown. We will show the results of higher corruption levels later.

In Figure 1, we show the cases of corruption level $C$ known and unknown to the learning agent. The bars in the plots are the variance of the data. For any non-robust algorithm, the regret rapidly increases with time, indicating that they rarely take the optimal arm during the whole learning process. We notice that the regret of the two vulnerable algorithms under the attack are almost identical. The reason is that the attacker highlights a target arm. For the vulnerable algorithms, they will believe that the target arm is optimal and almost always take that arm during training. So, their behavior and regret are very similar under the attack. The difference between the regrets of the two robust algorithms is subtle. After the first few timesteps, the regret increases slowly with time. This suggests that after some explorations, the robust algorithms successfully identify the optimal arm and take it for most of the time.

**Robustness evaluation and comparison under different corruption level:**

Here, we show the total regret of our algorithm under attacks at different corruption levels. For the baseline robust algorithm, we extend the standard UCB algorithm to its robust version, which we call 'Robust UCB (RUCB).' Due to the reward poisoning attacks with corruption level $C$, the half-width of arm $i$'s high probability confidence interval is increased by $C/k_i(t)$ where $k_i(t)$ is the number of times arm $i$ has been selected by $t$. The algorithm is parameterized by a robustness coefficient $\bar{C}$. When the corruption level $C$ is known to the agent, the agent can set $\bar{C} = C$ and use $\sqrt{2 \log T/k_i(t)} + C/k_i(t)$ as the exploration bonus to each arm. When the corruption level is unknown, the agent can set $\bar{C} = \beta \cdot \sqrt{T \log T/N}$ where $\beta$ is a constant, and use $\sqrt{\log T/k_i(t)} + \bar{C}/k_i(t)$ as the exploration bonus. This idea has also been mentioned in Lykouris et al. (2018). We show the RUCB algorithm in detail in the appendix. For both attack strategies we test with, the results in Figure 2 show that for the algorithms that are not robust, the regret becomes large quickly as the corruption level increases, indicating that these algorithms cannot find the optimal arm with even a low corruption level.

In the known corruption level case, the regret almost increases linearly with the corruption level for our robust algorithm, which agrees with our theoretical result. In the unknown corruption level case, we observe a threshold in the corruption level such that the regret of our algorithm increases rapidly with the corruption level, which is also aligned with our theoretical analysis. We notice that when the corruption level is small, the regret of our algorithm decreases slightly as the corruption level increases. The reason is that the attacker tries to highlight a randomly chosen target arm, and the arm is not the one with the lowest reward. Therefore, when the corruption level is not high enough to mislead the agent, the agent will only take sub-optimal arms for a limited number of rounds, and it tends to take the target arm more often instead of the worse arms, making the total regret slightly lower. The results also show that the performance of our robust Thompson sampling is similar to that of the robust UCB algorithm.

### 6.2 CONTEXTUAL LINEAR BANDIT SETTING

**Experiment setup:** We consider a linear contextual bandit environment with $N = 5$ arms and $T = 5000$ timesteps. The dimension of the context space is $d = 5$. For the baseline algorithms, we choose two standard algorithms, LinUCB and LinTS, to represent the vulnerable algorithms. They are the extensions of the UCB and Thompson sampling algorithm to the linear contextual case. We adopt two attack strategies: (1) Garcelon's attack proposed in Garcelon et al. (2020); (2) oracle MAB attack, which is also adopted in Garcelon et al. (2020).

**Cumulative regret during training under attack:** Here, we show the behavior of different learning algorithms under different attacks during the learning process. The corruption level for the attack is set as $C = 300$. In Figure 3, we show the cases of corruption level $C$ known and unknown to the

learning agent. Similar to the stochastic bandit case, the regret of the vulnerable algorithms rapidly increases with time, suggesting that they almost can never find the optimal arm. For our robust algorithms, after the first few timesteps, it learns to estimate the reward parameter accurately and can almost always find the optimal arm.

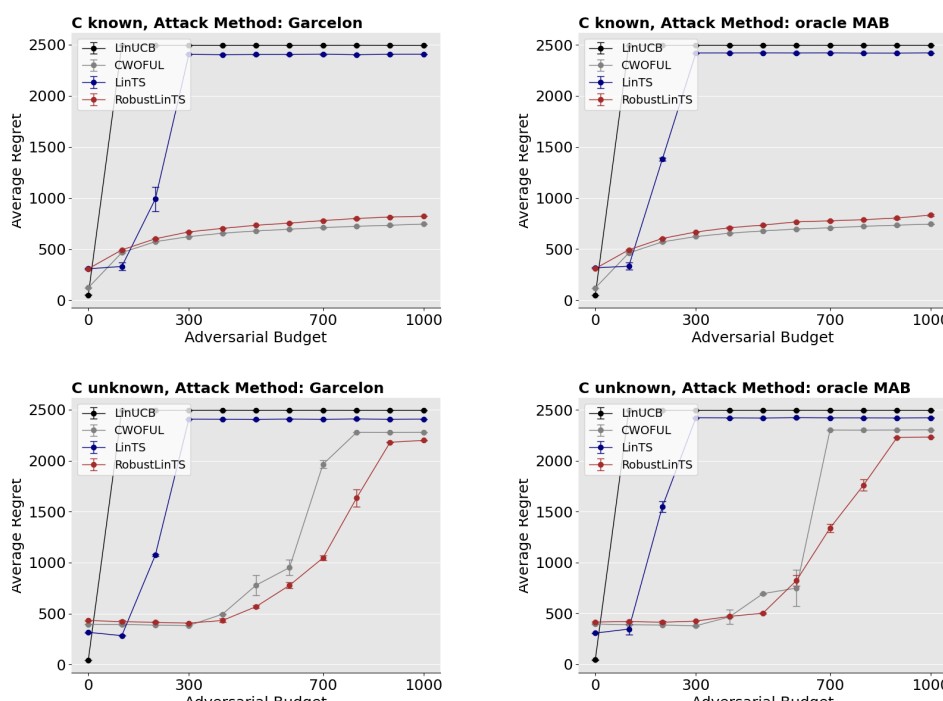

Figure 4: In the linear contextual bandit setting, the total regret of different algorithms under attacks at different corruption level.

**Robustness evaluation and comparison under different corruption level:** Here, we show the regret of different learning algorithms under attacks at different corruption levels. For comparison of robustness, We choose the state-of-the-art CW-OFUL algorithm He et al. (2022) as the robust baseline. For both attack strategies we test with, the results in Fig 4 show that for the vulnerable algorithms, the regret increases quickly as the corruption level increases and converges to a large value in the end, indicating that these algorithms can no longer find the optimal arm for even a relatively low corruption level.

For our robust algorithm, when $C$ is known to the agent, the regret increases linearly with the corruption level; when $C$ is unknown to the agent, there exists a threshold in the corruption level such that at one point, the regret rapidly increases with the corruption level. This observation agrees with our theoretical result. Figure 3 and 4 also show that our algorithm is as robust as the CW-OFUL algorithm. In our setup, our algorithm performs slightly worse in the known corruption level $C$ case and significantly better in the unknown $C$ case, especially when $C$ is large. Our robust algorithms not only inherit the idea of Thompson sampling exploration but also achieve state-of-the-art performance in practice.

## 7 CONCLUSION AND LIMITATION

In this work, we propose two robust Thompson sampling algorithms for stochastic and linear contextual MAB settings, with a theoretical guarantee of near-optimal regret. However, we mainly focus on the case where the posteriors of the arms are Gaussian distributions, though the theoretical analysis can be applied to other posterior settings. Our ideas for building robust Thompson sampling have been used in the two most popular bandit settings, and we do not cover settings like MDP. In the future, we aim to extend our techniques to other online learning settings.

## 8 REPRODUCIBILITY

We clearly explain the bandit settings and threat models we work on. The proofs for any Theorems and Lemmas can be found in the appendix. The codes we use for the simulations are included in the supplementary materials.

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

# A    PROOF FOR SECTION 4

## A.1    PROOF OF THEOREM 4.1

To begin with, it's easy to see that we only need to prove the case where $\overline{C} = C$ in Alg 5. Suppose we have set $\overline{C} = C$. Following the proof in Agrawal & Goyal (2017), first, we define two good events such that the agent is likely to pull the optimal arm when the events are true.

**Definition A.1** (Good Events)**.** For $i \neq 1$, define $E_i^\mu(t)$ is the event $\hat{\mu}_i(t) \leq \mu_i + \frac{\Delta_i}{3}$, and $E_i^\theta(t)$ is the event $\theta_i(t) \leq \mu_i - \frac{\Delta_i}{3}$. $\mu_i$, $\Delta_i$ are defined in Section 3.1.

$E_i^\mu(t)$ holds mean that the empirical post-attack mean of any sub-optimal arm is not much greater than its true mean, and $E_i^\theta(t)$ holds means that the sampled value of any sub-optimal arm is not much greater than its true mean. Intuitively, under such situations, the regret should be low.

Next, we define a random variable $p_{i,t}$ determined by $\mathcal{H}_{t-1}$. $p_{i,t}$ represents that for a history $\mathcal{H}_{t-1}$, the probability of the sample from the optimal arm's distribution being much higher than the means of other arms.

**Definition A.2.** Define, $p_{i,t}$ as the probability $p_{i,t} := \Pr\left(\theta_1(t) > \mu_1 - \frac{\Delta_i}{3} \mid \mathcal{H}_{t-1}\right)$.

We decompose the regret into different cases based on whether the good events are true or not. The following lemma bounds the expected number of arm pulls for arm $i$ when both the good events are true.

**Lemma A.3.**

$$\sum_{t=1}^{T} \Pr\left(i(t) = i, E_i^\mu(t), E_i^\theta(t)\right) \leq 72\left(e^{64} + 4\right) \frac{\ln\left(T\Delta_i^2\right)}{\Delta_i^2} + \frac{4}{\Delta_i^2}$$

Sampling from the optimistic posterior, the reward belief of the optimal arm is likely to be large even under poisoning attacks. Therefore, the value of $p_{i,t}$ is more likely to be large, and the probability of pulling any sub-optimal arm $i$ in this case will be small. We notice that the constant term here is a big value. In Section 6 we empirically show that in practice the constant term is small.

Next, we consider the case when only $E_i^\mu(t)$ is true. The key insight is that for a sub-optimal arm $i$, when the empirical post-attack mean $\hat{\mu}_i$ is close to the true mean $\mu_i$ and the arm has already been pulled many times, the probability that sampled $\theta_i(t)$ is large is low. As a result, the total number of times when the sub-optimal arm is pulled in this case is limited. The formal result is shown in A.4

**Lemma A.4.**

$$\sum_{t=1}^{T} \Pr\left(i(t) = i, \overline{E_i^\theta(t)}, E_i^\mu(t)\right)$$

$$\leq \sum_{t=1}^{T} \Pr\left(i(t) = i, \overline{E_i^\theta(t)}, \ E_i^\mu(t), k_i(t) \leq \max\{\frac{32\ln\left(T\Delta_i^2\right)}{\Delta_i^2}, \frac{12C}{\Delta_i}\}\right) + \frac{1}{\Delta_i^2}$$

At last, we consider the case when neither good event is true. The key insight is that when a sub-optimal arm $i$ has already been pulled many times, the probability that the empirical post-attack mean $\hat{\mu}_i$ is far from the true mean $\mu_i$ is low, and the total number of times it being pulled in this case is limited as shown in Lemma A.5.

**Lemma A.5.** *For $i \neq 1$,*

$$\sum_{t=1}^{T} \Pr\left(i(t) = i, \overline{E_i^\mu(t)}\right) \leq \sum_{t=1}^{T} \Pr\left(i(t) = i, \overline{E_i^\mu(t)}, k_i(t) \leq \max\{\frac{32\ln\left(T\Delta_i^2\right)}{\Delta_i^2}, \frac{12C}{\Delta_i}\}\right) + 1 + \frac{8}{\Delta_i^2}$$

Next, we can prove Theorem 4.1 by combining the lemmas.

*Proof of Theorem 4.1.* We decompose the expected number of plays of a suboptimal arm $i \neq 1$ depending on whether the good events hold or not.

$$\mathbb{E}\left[k_i(T)\right] = \sum_{t=1}^{T} \Pr(i(t) = i) = \sum_{t=1}^{T} \Pr\left(i(t) = i, E_i^{\mu}(t), E_i^{\theta}(t)\right) + \sum_{t=1}^{T} \Pr\left(i(t) = i, E_i^{\mu}(t), \overline{E_i^{\theta}(t)}\right) +$$

$$\sum_{t=1}^{T} \Pr\left(i(t) = i, \overline{E_i^{\mu}(t)}\right).$$

For each sub-optimal arm $i$, combine the bounds from Lemmas A.3 to A.5, we obtain

$$\mathbb{E}\left[k_i(T)\right] \leq 72\left(e^{64} + 4\right)\frac{\ln\left(T\Delta_i^2\right)}{\Delta_i^2} + \max\{\frac{12C}{\Delta_i}, \frac{32\ln\left(T\Delta_i^2\right)}{\Delta_i^2}\} + \frac{13}{\Delta_i^2} + 1$$

$\Rightarrow$ The total regret can be upper bouded by:

$$\mathbb{E}[\mathcal{R}(T)] = \sum_{i=1}^{N} \Delta_i \mathbb{E}\left[k_i(T)\right] \leq \sum_{i=1}^{N}[72\left(e^{64} + 5\right)\frac{\ln\left(T\Delta_i^2\right)}{\Delta_i} + 12C + \frac{13}{\Delta_i} + \Delta_i]$$

Then for every arm $i$ with $\Delta_i \geq e\sqrt{\frac{N \ln N}{T}}$, the expected regret is bounded by $O\left(\sqrt{NT \ln N} + NC + N\right)$. And for arms with $\Delta_i \leq e\sqrt{\frac{N \ln N}{T}}$, the total regret is bounded by $O(\sqrt{NT \ln N} + NC)$. By combining these results we prove the Theorem 4.1.

$\square$

## A.2 PROOF OF LEMMA A.3

**Lemma A.6** (Lemma 2.8 in Agrawal & Goyal (2017)). *For all $t, i \neq 1$ and all instantiations $H_{t-1}$ of $\mathcal{H}_{t-1}$ we have*

$$\Pr\left(i(t) = i, E_i^{\mu}(t), E_i^{\theta}(t) \mid H_{t-1}\right) \leq \frac{(1 - p_{i,t})}{p_{i,t}} \Pr\left(i(t) = 1, E_i^{\mu}(t), E_i^{\theta}(t) \mid H_{t-1}\right)$$

From the proof of Lemma 2.8 in Agrawal & Goyal (2017), it's not hard to find that it is the event $E_i^{\theta}(t)$ that holds the above inequality. Therefore, although the distribution of $\theta_i$ has been changed to make the algorithm robust against adversarial attack, their proof can still be applied directly to the lemma A.6.

**Lemma A.7.** *Let $s_j$ denote the time of the $j^{th}$ play of the first arm. Then,*

$$\mathbb{E}\left[\frac{1}{p_{i,s_j+1}} - 1\right] \leq \begin{cases} e^{64} + 5 \\ \frac{4}{T\Delta_i^2} & j > \frac{72\ln\left(T\Delta_i^2\right)}{\Delta_i^2} \end{cases}$$

*Proof.* Given $\mathcal{H}_{s_j}$, let $\Theta_j$ denote a random variable sampled from $\mathcal{N}\left(\hat{\mu}_1\left(s_j + 1\right) + \frac{C}{j+1}, \frac{1}{j+1}\right)$, and $\Theta_j^o$ denote a random variable sampled from $\mathcal{N}\left(\hat{\mu}_1^o\left(s_j + 1\right), \frac{1}{j+1}\right)$. We abbreviate $\hat{\mu}_1\left(s_j + 1\right)$ to $\hat{\mu}_1$ and $\hat{\mu}_1^o\left(s_j + 1\right)$ to $\hat{\mu}_1^o$ in the following. Let $G_j$ and $G_j^o$ be the geometric random variable denoting the number of consecutive independent trials until a sample of $\Theta_j$ or $\Theta_j^o$ becomes greater than $\mu_1 - \frac{\Delta_i}{3}$, respectively. Notice that $\hat{\mu}_1\left(s_j + 1\right) + \frac{C}{j+1} \geq \hat{\mu}_1^o\left(s_j + 1\right)$, then we have

$$p_{i,s_j+1} = \Pr\left(\Theta_j > \mu_1 - \frac{\Delta_i}{3} \mid \mathcal{H}_{s_j}\right) \geq \Pr\left(\Theta_j^o > \mu_1 - \frac{\Delta_i}{3} \mid \mathcal{H}_{s_j}\right)$$

$\Rightarrow$

$$\mathbb{E}\left[\frac{1}{p_{i,s_j+1}} - 1\right] \leq \mathbb{E}\left[\frac{1}{\Pr\left(\Theta_j^o > \mu_1 - \frac{\Delta_i}{3} \mid \mathcal{H}_{s_j}\right)} - 1\right]$$

From the result in Agrawal & Goyal (2017),

$$\mathbb{E}\left[\frac{1}{\Pr\left(\Theta_j^o > \mu_1 - \frac{\Delta_i}{3} \mid \mathcal{H}_{s_j}\right)} - 1\right] \leq \begin{cases} e^{64} + 5 \\ \frac{4}{T\Delta_i^2} \end{cases} \quad j > \frac{72\ln\left(T\Delta_i^2\right)}{\Delta_i^2}$$

Combining the above two inequalities, we derive the needed result. □

*Proof.* Let $s_k$ denote the time at which arm $i$ is pulled $k$ times. Set $s_0 = 0$. Now applying Lemma A.6 and Lemma A.7, we have

$$\sum_{t=1}^{T} \Pr\left(i(t) = i, E_i^{\mu}(t), E_i^{\theta}(t)\right) = \sum_{t=1}^{T} \mathbb{E}\left[\Pr\left(i(t) = i, E_i^{\mu}(t), E_i^{\theta}(t) \mid \mathcal{H}_{t-1}\right)\right]$$

$$\leq \sum_{t=1}^{T} \mathbb{E}\left[\frac{(1 - p_{i,t})}{p_{i,t}} \Pr\left(i(t) = 1, E_i^{\theta}(t), E_i^{\mu}(t) \mid \mathcal{H}_{t-1}\right)\right]$$

$$= \sum_{t=1}^{T} \mathbb{E}\left[\mathbb{E}\left[\frac{(1 - p_{i,t})}{p_{i,t}} I\left(i(t) = 1, E_i^{\theta}(t), E_i^{\mu}(t)\right) \mid \mathcal{H}_{t-1}\right]\right]$$

$$= \sum_{t=1}^{T} \mathbb{E}\left[\frac{(1 - p_{i,t})}{p_{i,t}} I\left(i(t) = 1, E_i^{\theta}(t), E_i^{\mu}(t)\right)\right]$$

$$= \sum_{k=0}^{T-1} \mathbb{E}\left[\frac{(1 - p_{i,s_k+1})}{p_{i,s_k+1}} \sum_{t=s_k+1}^{s_{k+1}} I\left(i(t) = 1, E_i^{\theta}(t), E_i^{\mu}(t)\right)\right]$$

$$\leq \sum_{k=0}^{T-1} \mathbb{E}\left[\frac{(1 - p_{i,s_k+1})}{p_{i,s_k+1}}\right]$$

$$\leq 72\left(e^{64} + 4\right)\right) \frac{\ln\left(T\Delta_i^2\right)}{\Delta_i^2} + \frac{4}{\Delta_i^2}$$

Then we obtain the bound in Lemma A.3. □

### A.3 PROOF OF LEMMA A.4

*Proof.* We have

$$\sum_{t=1}^{T} \Pr\left(i(t) = i, \overline{E_i^{\theta}(t)}, E_i^{\mu}(t)\right) = \sum_{t=1}^{T} \Pr\left(i(t) = i, k_i(t) \leq \max\{\frac{32\ln\left(T\Delta_i^2\right)}{\Delta_i^2}, \frac{12C}{\Delta_i}\}, \overline{E_i^{\theta}(t)}, E_i^{\mu}(t)\right) +$$

$$\sum_{t=1}^{T} \Pr\left(i(t) = i, k_i(t) > \max\{\frac{32\ln\left(T\Delta_i^2\right)}{\Delta_i^2}, \frac{12C}{\Delta_i}\}, \overline{E_i^{\theta}(t)}, E_i^{\mu}(t)\right)$$

Next, we prove that the probability that the event $E_i^{\theta}(t)$ is violated is small when $k_i(t)$ is large enough and $E_i^{\mu}(t)$ holds. Notice that

$$\sum_{t=1}^{T} \Pr\left(i(t) = i, k_i(t) > \max\{\frac{32\ln\left(T\Delta_i^2\right)}{\Delta_i^2}, \frac{12C}{\Delta_i}\}, \overline{E_i^{\theta}(t)}, E_i^{\mu}(t)\right)$$

$$\leq \mathbb{E}\left[\sum_{t=1}^{T} \Pr\left(i(t) = i, \overline{E_i^{\theta}(t)} \mid k_i(t) > \max\{\frac{32\ln\left(T\Delta_i^2\right)}{\Delta_i^2}, \frac{12C}{\Delta_i}\}, E_i^{\mu}(t), \mathcal{H}_{t-1}\right)\right]$$

$$\leq \mathbb{E}\left[\sum_{t=1}^{T} \Pr\left(\theta_i(t) > \mu_1 - \frac{\Delta_i}{3} \mid k_i(t) > \max\{\frac{32\ln\left(T\Delta_i^2\right)}{\Delta_i^2}, \frac{12C}{\Delta_i}\}, E_i^{\mu}(t), \mathcal{H}_{t-1}\right)\right]$$

Recall that $\theta_i(t)$ is sampled from $\mathcal{N}\left(\hat{\mu}_i(t) + \frac{C}{k_i(t)+1}, \frac{1}{k_i(t)+1}\right)$. Since $\hat{\mu}_i(t) \leq \mu_i + \frac{\Delta_i}{3}$, we have

$$\Pr\left(\theta_i(t) > \mu_1 - \frac{\Delta_i}{3} \mid k_i(t) > \max\{\frac{32\ln\left(T\Delta_i^2\right)}{\Delta_i^2}, \frac{12C}{\Delta_i}\}, \hat{\mu}_i(t) \leq \mu_i + \frac{\Delta_i}{3}, \mathcal{H}_{t-1}\right)$$

$$\leq \Pr\left(\mathcal{N}\left(\mu_i + \frac{\Delta_i}{3} + \frac{C}{k_i(t)+1}, \frac{1}{k_i(t)+1}\right) > \mu_1 - \frac{\Delta_i}{3} \mid \mathcal{H}_{t-1}, k_i(t) > \max\{\frac{32\ln\left(T\Delta_i^2\right)}{\Delta_i^2}, \frac{12C}{\Delta_i}\}\right)$$

Since $k_i(t) > \max\{\frac{32\ln\left(T\Delta_i^2\right)}{\Delta_i^2}, \frac{12C}{\Delta_i}\}$, from the property of Gaussian distribution we obtain that

$$\Pr\left(\mathcal{N}\left(\mu_i + \frac{\Delta_i}{3} + \frac{C}{k_i(t)+1}, \frac{1}{k_i(t)+1}\right) > \mu_1 - \frac{\Delta_i}{3}\right) \leq \frac{1}{2}e^{-\frac{(k_i(t)+1)\left(\frac{\Delta_i}{3} - \frac{C}{k_i(t)+1}\right)^2}{2}}$$

$$\leq \frac{1}{2}e^{-\frac{(k_i(t)+1)\left(\frac{\Delta_i}{3}\right)^2}{2}}$$

$$\leq \frac{1}{T\Delta_i^2}$$

the second inequality holds because $k_i(t) \geq \frac{12C}{\Delta_i}$ and the last inequality holds because $k_i(t) \geq \frac{32\ln\left(T\Delta_i^2\right)}{(\Delta_i)^2}$. Therefore,

$$\Pr\left(\theta_i(t) > \mu_1 - \frac{\Delta_i}{3} - \frac{\Delta_i}{12} \mid k_i(t) > \max\{\frac{32\ln\left(T\Delta_i^2\right)}{\Delta_i^2}, \frac{12C}{\Delta_i}\}, \hat{\mu}_i(t) \leq \mu_i + \frac{\Delta_i}{3}, \mathcal{H}_{t-1}\right) \leq \frac{1}{T\Delta_i^2}$$

$$\Rightarrow$$

$$\sum_{t=1}^T \Pr\left(i(t) = i, k_i(t) > \max\{\frac{32\ln\left(T\Delta_i^2\right)}{\Delta_i^2}, \frac{12C}{\Delta_i}\}, \overline{E_i^\theta(t)}, E_i^\mu(t)\right) \leq \frac{1}{\Delta_i^2}$$

Then we finish the proof. $\qquad\square$

## A.4 PROOF OF LEMMA A.5

*Proof.* Let $s_k$ denote the time at which arm $i$ is pulled $k$ times. Set $s_0 = 0$. We have

$$\sum_{t=1}^T \Pr\left(i(t) = i, \overline{E_i^\mu(t)}\right) \leq \sum_{t=1}^T \Pr\left(i(t) = i, \overline{E_i^\mu(t)}, k_i(t) \leq \max\{\frac{32\ln\left(T\Delta_i^2\right)}{\Delta_i^2}, \frac{12C}{\Delta_i}\}\right) +$$

$$\sum_{k > \max\{\frac{32\ln\left(T\Delta_i^2\right)}{\Delta_i^2}, \frac{12C}{\Delta_i}\}}^{T-1} \Pr\left(\overline{E_i^\mu\left(s_{k+1}\right)}\right)$$

At time $s_{k+1}$ for $k \geq 1$, we have $\hat{\mu}_i\left(s_{k+1}\right) = \frac{\sum_{t=1, i(t)=i}^{s_{k+1}} r_i(t)}{k+1} \leq \frac{\sum_{t=1, i(t)=i}^{s_{k+1}} r_i^o(t)}{k+1} + \frac{C}{k+1}$. By Chernoff-Hoeffding inequality, when $k > \max\{\frac{32\ln\left(T\Delta_i^2\right)}{\Delta_i^2}, \frac{12C}{\Delta_i}\}$,

$$\Pr\left(\hat{\mu}_i\left(s_{k+1}\right) > \mu_i + \frac{\Delta_i}{3}\right)$$

$$\leq \Pr\left(\frac{\sum_{t=1, i(t)=i}^{s_{k+1}} r_i^o(t)}{k+1} + \frac{C}{k+1} > \mu_i + \frac{\Delta_i}{3}\right) \leq e^{-2(k+1)\left(\frac{\Delta_i}{3} - \frac{C}{k+1}\right)^2} \leq e^{\frac{-(k+1)\Delta_i^2}{8}}$$

we then obtain that

$$\sum_{k > \max\{\frac{32\ln\left(T\Delta_i^2\right)}{\Delta_i^2}, \frac{12C}{\Delta_i}\}}^{T-1} \Pr\left(\overline{E_i^\mu\left(s_{k+1}\right)}\right) = \sum_{k > \max\{\frac{32\ln\left(T\Delta_i^2\right)}{\Delta_i^2}, \frac{12C}{\Delta_i}\}}^{T-1} \Pr\left(\hat{\mu}_i\left(s_{k+1}\right) > \mu_i + \frac{\Delta_i}{3}\right)$$

$$\leq 1 + \sum_{k=1}^{T-1} \exp\left(-\frac{(k+1)\Delta_i^2}{8}\right) \leq 1 + \frac{8}{\Delta_i^2}$$

Combining the above results we finish the proof. $\qquad\square$

## B  PROOF FROM SECTION 5

### B.1  PROOF OF THEOREM 5.1

Follow the proof of Agrawal & Goyal (2013), to prove Theorem 5.1, we begin with some basic definitions. The sample $\tilde{\mu}(t)$ from the posterior is a belief in the reward parameter. Therefore, we denote the actual sample for the belief in the reward of an arm as $\theta_i(t) := x_i(t)^T \tilde{\mu}(t)$. By definition of $\tilde{\mu}(t)$ in Algorithm 2, marginal distribution of each $\theta_i(t)$ is Gaussian with mean $x_i(t)^T \hat{\mu}(t)$ and standard deviation $v_t \|x_i(t)\|_{B(t)^{-1}}$. Similar to before, we denote $\Delta_i(t) := x_{i^*(t)}(t)^T \mu - x_i(t)^T \mu$ as the gap between the mean reward of optimal arm and arm $i$ at time $t$, and we define two good events as Definition 5.2

Next, we define a notion called saturated arm to indicate whether an arm has been taken for enough time such that the variance in its reward estimation is less than the gap in reward compared to the optimal arm at a timestep.

**Definition B.1** (Saturated Arm). Denote $g_t = \sqrt{4d \ln(t)} v_t + \sigma \sqrt{d \ln\left(\frac{t^3}{\delta}\right)} + 1 + C\gamma$. An arm $i$ is called saturated at time $t$ if $\Delta_i(t) > g_t \|x_i(t)\|_{B(t)^{-1}}$, and unsaturated otherwise. Let $C(t)$ ne the set of saturated arms at time $t$.

First, in Lemma B.2 we show that good events hold with a high probability at each round. The reason why Lemma B.2 holds is because of the weighted ridge estimator we use for computing the posteriors. With such an estimator, the attack has less influence on the estimations, and therefore the difference between the estimation and the true value can be bounded.

**Lemma B.2.** *For all $t, 0 < \delta < 1, \Pr\left(E^\mu(t)\right) \geq 1 - \frac{\delta}{t^2}$. For all possible filtrations $\mathcal{H}_{t-1}, \Pr\left(E^\theta(t) \mid \mathcal{H}_{t-1}\right) \geq 1 - \frac{1}{t^2}$.*

Next, Lemma B.3 shows that when the good events are true, with a high probability, the sampled reward for the optimal arm at a timestep is likely to be larger than its actual expected reward.

**Lemma B.3.** *Denote $p = \frac{1}{4e^{\left(1 + \frac{C\gamma}{\sqrt{d}}\right)^2} \sqrt{\pi}}$. For any filtration $\mathcal{H}_{t-1}$ such that $E^\mu(t)$ is true,*

$$\Pr\left(\theta_{i^*(t)}(t) > x_{i^*(t)}(t)^T \mu \mid \mathcal{H}_{t-1}\right) \geq p$$

Lemma B.3 suggests that the algorithm is unlikely to underestimate the reward of the optimal arm.

Based on Lemma B.2 and Lemma B.3, we can further show that when the good events are true, since the optimal arm and the unsaturated arm will be pulled with at least a certain probability, the algorithm can perform effective exploration. Therefore, the regret will be upper bounded with a high probability. We establish a super-martingale process that will form the basis of our proof of the high-probability regret bound. This result shows that expected regret will be $O\left(\frac{g_t}{p} * \sum_t \|x_{i(t)}(t)\|_{B(t)^{-1}}\right)$.

**Definition B.4.** Recall that regret $(t)$ was defined as, regret $(t) = \Delta_{i(t)}(t) = x_{i^*(t)}(t)^T \mu - x_{i(t)}(t)^T \mu$. Define regret $'(t) = \text{regret}(t) \cdot I\left(E^\mu(t)\right)$.

**Definition B.5.** Let

$$X_t = \text{regret}'(t) - \min\left\{\frac{3g_t}{p}\|x_{i(t)}(t)\|_{B(t)^{-1}} + \frac{2g_t}{pt^2}, 1\right\}$$

$$Y_t = \sum_{w=1}^{t} X_w.$$

**Lemma B.6.** $(Y_t; t = 0, \ldots, T)$ *is a super-martingale process with respect to filtration $\mathcal{H}_t$.*

*Proof of Theorem 5.1.* Note that $X_t$ is bounded, $|X_t| \leq 1 + \frac{3}{p}g_t + \frac{2}{pt^2}g_t \leq \frac{6}{p}g_t$. Thus, we can apply the Azuma-Hoeffding inequality, to obtain that with probability $1 - \frac{\delta}{2}$,

$$\sum_{t=1}^{T} \text{regret}'(t) \leq \sum_{t=1}^{T} \min\left\{\frac{3g_t}{p} s_{a(t)}(t), 1\right\} + \sum_{t=1}^{T} \frac{2g_t}{pt^2} + \sqrt{2\left(\sum_t \frac{36g_t^2}{p^2}\right) \ln\left(\frac{2}{\delta}\right)}$$

Note that $p$ is a constant. Also, by definition, $g_t \leq g_T$. Therefore, from above equation, with probability $1 - \frac{\delta}{2}$,

$$\sum_{t=1}^{T} \text{regret}'(t) \leq \sum_{t=1}^{T} \min\{\frac{3g_t}{p} s_{i(t)}(t), 1\} + \frac{2g_T}{p} \sum_{t=1}^{T} \frac{1}{t^2} + \frac{6g_T}{p}\sqrt{2T \ln\left(\frac{2}{\delta}\right)}$$

Now, we have

$$\sum_{t=1}^{T} \min\{1, \frac{3g_k}{p} s_{t,i(t)}\}$$

$$= \underbrace{\sum_{k:w_k=1} \min\left(1, \frac{3g_t}{p}\sqrt{x_{i(k)}(k)^\top B_{i(k)}^{-1} x_{i(k)}(k)}\right)}_{I_1} + \underbrace{\sum_{k:w_k<1} \min\left(1, \frac{3g_k}{p}\sqrt{x_{i(k)}(k)^\top B_{i(k)}^{-1} x_{i(k)}(k)}\right)}_{I_2},$$

For the term $I_1$, we consider all rounds $k \in [T]$ with $w_k = 1$ and we assume these rounds can be listed as $\{k_1, \cdots, k_m\}$ for simplicity. With this notation, for each $i \leq m$, we can construct the auxiliary covariance matrix $A_i = \lambda I + \sum_{j=1}^{i-1} x_{i(k_j)}(k_j) x_{i(k_j)}(k_j)^\top$. Due to the definition of original covariance matrix $B_k$ in Algorithm, we have

$$B_{k_i} \geq \lambda I + \sum_{j=1}^{i-1} w_{k_j} x_{i(k_j)}(k_j) x_{i(k_j)}(k_j)^\top = A_i$$

It further implies that for vector $x_{k_i}$, we have

$$x_{i(k)}(k)^\top B_{i(k)}^{-1} x_{i(k)}(k) \leq x_{i(k)}(k)^\top A_i^{-1} x_{i(k)}(k)$$

The term $I_1$ can be bounded by

$$I_1 = \sum_{k:w_k=1} \min\left(1, \frac{3g_t}{p}\sqrt{x_{i(k)}(k)^\top B_{i(k)}^{-1} x_{i(k)}(k)}\right)$$

$$\leq \sum_{i=1}^{m} \frac{3g_t}{p} \min\left(1, \sqrt{x_{i(k_i)}(k_i)^\top A_i^{-1} x_{i(k_i)}(k_i)}\right)$$

$$\leq \frac{3g_t}{p}\sqrt{\sum_{i=1}^{m} 1 \times \sum_{i=1}^{m} \min\left(1, x_{i(k_i)}(k_i)^\top A_i^{-1} x_{i(k_i)}(k_i)\right)}$$

$$\leq \frac{3g_t}{p}\sqrt{2dT \ln(1+T)},$$

For the second term $I_2$, according to the definition for weight $w_k < 1$ in Algorithm 1, we have $w_k = \gamma/\sqrt{x_{i(k)}(k)^\top B_k^{-1} x_{i(k)}(k)}$, which implies that

$$I_2 = \sum_{k:w_k<1} \min\left(1, \frac{3g_t}{p}\sqrt{x_{i(k)}(k)^\top B_{i(k)}^{-1} x_{i(k)}(k)}\right)$$

$$= \sum_{k:w_k<1} \min\left(1, \frac{3g_t}{p} w_k x_{i(k)}(k)^\top B_{i(k)}^{-1} x_{i(k)}(k)/\gamma\right)$$

$$\leq \sum_{k:w_k<1} \min\left((1+\frac{3g_t}{p\gamma}), (1+\frac{3g_t}{p\gamma}) w_k x_{i(k)}(k)^\top B_{i(k)}^{-1} x_{i(k)}(k)\right)$$

$$= \sum_{k:w_k<1} (1+\frac{3g_t}{p\gamma}) \min\left(1, w_k x_{i(k)}(k)^\top B_{i(k)}^{-1} x_{i(k)}(k)\right)$$

where the first equation holds due to the definition of weight $w_k$. Now, we assume the rounds with weight $w_k < 1$ can be listed as $\{k_1, \cdots, k_m\}$ for simplicity. In addition, we introduce the auxiliary vector $x'_i$ as $x'_i = \sqrt{w_{k_i}} x_{i(k_i)}(k_i)$ and matrix $B'_i$ as

$$B'_i = \lambda I + \sum_{j=1}^{i-1} w_{k_j} x_{i(k_j)}(k_j) x_{i(k_j)}(k_j)^\top = \lambda I + \sum_{j=1}^{i-1} x'_j \left(x'_j\right)^\top.$$

We have $\left(B'_i\right)^{-1} \succeq B_i^{-1}$. Therefore, for each $i \in [m]$, we have

$$x_{i(k_i)}(k_i)^\top \left(B'_i\right)^{-1} x_{i(k_i)}(k_i) \geq x_{i(k_i)}(k_i)^\top B_i^{-1} x_{i(k_i)}(k_i)$$

Now we have

$$\sum_{i=1}^{m} \min\left(1, w_{k_i} x_{i(k_i)}(k_i)^\top B_{i(k_i)}^{-1} x_{i(k_i)}(k_i)\right)$$

$$\leq \sum_{i=1}^{m} \min\left(1, w_{k_i} x_{i(k_i)}(k_i)^\top \left(B'_i\right)^{-1} x_{i(k_i)}(k_i)\right)$$

$$= \sum_{i=1}^{m} \min\left(1, \left(x'_i\right)^\top \left(B'_i\right)^{-1} x'_i\right)$$

$$\leq 2d \ln\left(1 + T\right),$$

Then we have

$$I_2 \leq \sum_{k:w_k < 1} (2 + \frac{3g_t}{p\gamma}) \min\left(1, w_k x_{i(k_i)}(k_i)^\top B_{i(k_i)}^{-1} x_{i(k_i)}(k_i)\right)$$

$$\leq 2d(1 + \frac{3g_t}{p\gamma}) \ln\left(1 + T\right).$$

Recalling the definitions of $p$ and $g_T$, by definition $g_T = O\left(d\sqrt{\ln\left(\frac{T}{\delta}\right)} + C\gamma\right)$. Substituting the above, we get

$$\sum_{t=1}^{T} \text{regret}'(t)$$

$$= O\left(e^{(1+\frac{C\gamma}{\sqrt{d}})^2}\left(d\sqrt{\ln\left(\frac{T}{\delta}\right)} + C\gamma\right) \cdot \sqrt{dT \ln T} + e^{(1+\frac{C\gamma}{\sqrt{d}})^2}\left(d\sqrt{\ln\left(\frac{T}{\delta}\right)} + C\gamma\right)\frac{d}{\gamma}\ln T + 2d\ln T\right)$$

$$= O\left(de^{(1+\frac{C\gamma}{\sqrt{d}})^2}\sqrt{dT \ln T \ln\left(\frac{T}{\delta}\right)} + C\gamma e^{(1+\frac{C\gamma}{\sqrt{d}})^2}\sqrt{dT \ln T} + \frac{d^2 e^{(1+\frac{C\gamma}{\sqrt{d}})^2}}{\gamma}\ln T\sqrt{\ln\left(\frac{T}{\delta}\right)} + \right.$$

$$\left. Cde^{(1+\frac{C\gamma}{\sqrt{d}})^2}\ln T\right)$$

Also, because $E^\mu(t)$ holds for all $t$ with probability at least $1 - \frac{\delta}{2}$ (see Lemma B.2), $\text{regret}'(t) = \text{regret}(t)$ for all $t$ with probability at least $1 - \frac{\delta}{2}$. Hence, with probability $1 - \delta$,

$$\mathcal{R}(T) = \sum_{t=1}^{T} \text{regret}(t) = \sum_{t=1}^{T} \text{regret}'(t)$$

$$= O\left(de^{(1+\frac{C\gamma}{\sqrt{d}})^2}\sqrt{dT \ln T \ln\left(\frac{T}{\delta}\right)} + C\gamma e^{(1+\frac{C\gamma}{\sqrt{d}})^2}\sqrt{dT \ln T} + \frac{d^2 e^{(1+\frac{C\gamma}{\sqrt{d}})^2}}{\gamma}\ln T\sqrt{\ln\left(\frac{T}{\delta}\right)}\right.$$

$$\left. + Cde^{(1+\frac{C\gamma}{\sqrt{d}})^2}\ln T\right).$$

Choose $\gamma = \sqrt{d}/C$, its regret can be upper bounded by $\mathcal{R}(T) = O\left(d\sqrt{dT \ln T \ln\left(\frac{T}{\delta}\right)} + Cd\sqrt{d}\ln T\sqrt{\ln\left(\frac{T}{\delta}\right)}\right).$

$\square$

### B.2 PROOF OF LEMMA B.2

*Proof.* First, the probability bound for $E^\mu(t)$ can be directly obtained from Lemma 1 in Agrawal & Goyal (2013).

Now we bound the probability of event $E^\mu(t)$. We use Lemma C.3 with $m_t = \sqrt{w_t}x_{i(t)}(t), \epsilon_t = \sqrt{w_t}(r^0_{i(t)}(t) - x_{i(t)}(t)^T\mu), \mathcal{H}'_t = (a(s+1), m_{s+1}, \epsilon_s : s \le t)$. By the definition of $\mathcal{H}'_t$, $m_t$ is $\mathcal{H}'_{t-1}$-measurable, and $\epsilon_t$ is $\mathcal{H}'_t$-measurable. $\epsilon_t$ is conditionally $\sigma$-sub-Gaussian due to $\sqrt{w_t} \le 1$ and the problem setting, and is a martingale difference process:

$$\mathbb{E}\left[\sqrt{w_t}\epsilon_t \mid \mathcal{H}'_{t-1}\right] = \mathbb{E}\left[\sqrt{w_t}r^0_{i(t)}(t) \mid x_{i(t)}(t), i(t)\right] - \sqrt{w_t}x_{i(t)}(t)^T\mu = 0$$

We denote

$$M_t = I_d + \sum_{s=1}^{t} m_s m_s^T = I_d + \sum_{s=1}^{t} w_t x_{i(s)}(s) x_{i(s)}(s)^T$$

$$\xi_t = \sum_{s=1}^{t} m_s \epsilon_s = \sum_{s=1}^{t} w_t x_{i(s)}(s)\left(r^0_{i(s)} - x_{i(s)}(s)^T\mu\right)$$

Note that $B(t) = M_{t-1}$, and $\hat{\mu}(t) - \mu = M_{t-1}^{-1}\left(\xi_{t-1} - \mu + \sum_{s=1}^{t} w_s x_{i(s)}(s)c(s)\right)$. Let for any vector $y \in \mathbb{R}$ and matrix $A \in \mathbb{R}^{d \times d}$, $\|y\|_A$ denote $\sqrt{y^T A y}$. Then, for all $i$,

$$\left|x_i(t)^T\hat{\mu}(t) - x_i(t)^T\mu\right| = \left|x_i(t)^T M_{t-1}^{-1}\left(\xi_{t-1} - \mu + \sum_{s=1}^{t} w_s x_{i(s)}(s)c(s)\right)\right|$$

$$\le \|x_i(t)\|_{M_{t-1}^{-1}}\|\xi_{t-1} - \mu + \sum_{s=1}^{t} w_s x_{i(s)}(s)c(s)\|_{M_{t-1}^{-1}}$$

$$\le (\|\xi_{t-1} - \mu\|_{M_{t-1}^{-1}} + \sum_{s=1}^{t} |w_s||c(s)|\|x_{i(s)}(s)\|_{B(t)^{-1}})$$

$$\|x_i(t)\|_{B(t)^{-1}}$$

The inequality holds because $M_{t-1}^{-1}$ is a positive definite matrix. Using Lemma C.3 , for any $\delta' > 0, t \ge 1$, with probability at least $1 - \delta'$,

$$\|\xi_{t-1}\|_{M_{t-1}^{-1}} \le \sigma\sqrt{d\ln\left(\frac{t}{\delta'}\right)}$$

Therefore, $\|\xi_{t-1} - \mu\|_{M_{t-1}^{-1}} \le R\sqrt{d\ln\left(\frac{t}{\delta'}\right)} + \|\mu\|_{M_{t-1}^{-1}} \le R\sqrt{d\ln\left(\frac{t}{\delta'}\right)} + 1$. By the definition of $w_k$, we also note that $\sum_{s=1}^{t} |w_s||c(s)|\|x_{i(s)}(s)\|_{B_t^{-1}} \le \gamma\sum_{s=1}^{t} |c(s)| \le C\gamma$. Substituting $\delta' = \frac{\delta}{t^2}$, we get that with probability $1 - \frac{\delta}{t^2}$, for all $i$,

$$\left|x_i(t)^T\hat{\mu}(t) - x_i(t)^T\mu\right| \le \|x_i(t)\|_{B(t)^{-1}} \cdot \left(R\sqrt{d\ln\left(\frac{t}{\delta'}\right)} + 1 + C\gamma\right)$$

$$\le \|x_i(t)\|_{B(t)^{-1}} \cdot \left(R\sqrt{d\ln(t^3)\ln\left(\frac{1}{\delta}\right)} + 1 + C\gamma\right)$$

$$= \ell(t)s_i(t).$$

This proves the bound on the probability of $E^\mu(t)$. $\square$

### B.3 PROOF OF LEMMA B.3

*Proof.* Given event $E^\mu(t)$, $\left|x_{i^*(t)}(t)^T\hat{\mu}(t) - x_{i^*(t)}(t)^T\mu\right| \le \ell_t s_{t,i^*(t)}(t)$. And, since Gaussian random variable $\theta_{i^*(t)}(t)$ has mean $x_{i^*(t)}(t)^T\hat{\mu}(t)$ and standard deviation $v_t s_{i^*(t)}(t)$, using Lemma

C.1,

$$\Pr\left(\theta_{i^*(t)}(t) \geq x_{i^*(t)}(t)^T \mu \mid \mathcal{H}_{t-1}\right)$$

$$= \Pr\left(\frac{\theta_{i^*(t)}(t) - x_{i^*(t)}(t)^T \hat{\mu}(t)}{v_t s_{t,i^*(t)}} \geq \frac{x_{i^*(t)}(t)^T \mu - x_{i^*(t)}(t)^T \hat{\mu}(t)}{v_t s_{t,i^*(t)}} \mid \mathcal{H}_{t-1}\right)$$

$$\geq \frac{1}{4\sqrt{\pi}} e^{-Z_t^2}$$

where

$$|Z_t| = \left| \frac{x_{i^*(t)}(t)^T \mu - x_{i^*(t)}(t)^T \hat{\mu}(t)}{v_t s_{i^*(t)}(t)} \right|$$

$$\leq \frac{\ell_t s_{i^*(t)}(t)}{v_t s_{i^*(t)}(t)}$$

$$= \frac{\left(\sigma\sqrt{d \ln\left(\frac{t^3}{\delta}\right)} + 1 + C\gamma\right)}{\sigma\sqrt{9d \ln\left(\frac{t}{\delta}\right)}}$$

$$\leq 1 + \frac{C\gamma}{\sqrt{d}}$$

Therefore

$$\Pr\left(\theta_{i^*(t)}(t) \geq x_{i^*(t)}(t)^T \mu \mid \mathcal{H}_{t-1}\right) \geq \frac{1}{4 e^{(1 + \frac{C\gamma}{\sqrt{d}})^2} \sqrt{\pi}}$$

$\square$

## B.4 PROOF OF LEMMA B.6

**Lemma B.7.** *For any filtration $\mathcal{H}_{t-1}$ such that $E^\mu(t)$ is true,*

$$\Pr\left(i(t) \notin C(t) \mid \mathcal{H}_{t-1}\right) \geq p - \frac{1}{t^2}.$$

This Lemma is from Lemma 4 in Agrawal & Goyal (2013). By using the Lemma B.2, we get that when both events $E^\mu(t)$ and $E^\theta(t)$ hold, for all $j \in C(t)$, $\theta_j(t) \leq b_j(t)^T \mu + g_t s_{t,j}$. Also, by Lemma B.3, we have that if $E^\mu(t)$ is true, $\Pr\left(\theta_{i^*(t)}(t) > x_{i^*(t)}(t)^T \mu \mid \mathcal{H}_{t-1}\right) \geq p$. Then directly following the proof of Lemma 3 in Agrawal & Goyal (2013) we can obtain our result.

**Lemma B.8.** *For any filtration $\mathcal{H}_{t-1}$ such that $E^\mu(t)$ is true,*

$$\mathbb{E}\left[\Delta_{i(t)}(t) \mid \mathcal{H}_{t-1}\right] \leq \min\{\frac{3g_t}{p} \mathbb{E}\left[s_{i(t)}(t) \mid \mathcal{H}_{t-1}\right] + \frac{2g_t}{pt^2}, 1\}$$

This Lemma is from Lemma 4 in Agrawal & Goyal (2013). Using Lemma B.7, for any $\mathcal{H}_{t-1}$ such that $E^\mu(t)$ is true, we have $\Pr\left(i(t) \notin C(t) \mid \mathcal{H}_{t-1}\right) \geq p - \frac{1}{t^2} = \frac{1}{4 e^{(1 + \frac{C\gamma}{\sqrt{d}})^2} \sqrt{\pi}} - \frac{1}{t^2}$. Also, by Lemma B.2 we have that on the events $E^\mu(t)$ and $E^\theta(t)$, $\theta_i(t) \leq x_i(t)^T \mu + g_t \|x_i(t)\|_{B(t)^{-1}}$. Using these two facts, by directly following the proof in Lemma 4 of Agrawal & Goyal (2013) we immediately obtain our needed result.

*Proof.* We need to prove that for all $t \in [1, T]$, and any $\mathcal{H}_{t-1}$, $\mathbb{E}\left[Y_t - Y_{t-1} \mid \mathcal{H}_{t-1}\right] \leq 0$, i.e.

$$\mathbb{E}\left[\text{regret}'(t) \mid \mathcal{H}_{t-1}\right] \leq \min\{\frac{3g_t}{p} \mathbb{E}\left[s_{i(t)}(t) \mid \mathcal{H}_{t-1}\right] + \frac{2g_t}{pt^2}, 1\}$$

Note that whether $E^\mu(t)$ is true or not is completely determined by $\mathcal{H}_{t-1}$. If $\mathcal{H}_{t-1}$ is such that $E^\mu(t)$ is not true, then $\text{regret}'(t) = \text{regret}(t) \cdot I\left(E^\mu(t)\right) = 0$, and the above inequality holds trivially. And, for $\mathcal{H}_{t-1}$ such that $E^\mu(t)$ holds, the inequality follows from Lemma B.8. $\square$

## C  INEQUALITIES

**Lemma C.1.** *For a Gaussian distributed random variable $Z$ with mean $m$ and variance $\sigma^2$, for any $z \geq 1$,*

$$\frac{1}{2\sqrt{\pi}z}e^{-z^2/2} \leq \Pr(|Z - m| > z\sigma) \leq \frac{1}{\sqrt{\pi}z}e^{-z^2/2}.$$

**Lemma C.2** (Azuma-Hoeffding inequality). *If a super-martingale $(Y_t; t \geq 0)$, corresponding to filtration $\mathcal{H}_t$, satisfies $|Y_t - Y_{t-1}| \leq C_t$ for some constant $C_t$, for all $t = 1, \ldots, T$, then for any $a \geq 0$,*

$$\Pr(Y_T - Y_0 \geq a) \leq e^{-\frac{a^2}{2\sum_{t=1}^{T} C_t^2}}$$

**Lemma C.3** (Abbasi-Yadkori et al. (2011)). *Let $(\mathcal{H}'_t; t \geq 0)$ be a filtration, $(m_t; t \geq 1)$ be an $\mathbb{R}^d$-valued stochastic process such that $m_t$ is $(\mathcal{H}'_{t-1})$-measurable, $(\eta_t; t \geq 1)$ be a real-valued martingale difference process such that $\eta_t$ is $(\mathcal{H}'_t)$-measurable. For $t \geq 0$, define $\xi_t = \sum_{\tau=1}^{t} m_\tau \eta_\tau$ and $M_t = I_d + \sum_{\tau=1}^{t} m_\tau m_\tau^T$, where $I_d$ is the d-dimensional identity matrix. Assume $\eta_t$ is conditionally $R$-sub-Gaussian. Then, for any $\delta' > 0, t \geq 0$, with probability at least $1 - \delta'$,*

$$\|\xi_t\|_{M_t^{-1}} \leq R\sqrt{d \ln\left(\frac{t+1}{\delta'}\right)},$$

*where $\|\xi_t\|_{M_t^{-1}} = \sqrt{\xi_t^T M_t^{-1} \xi_t}$.*

## D  ALGORITHMS

---
**Algorithm 3** Thompson Sampling for Stochastic Bandits
---
1: For each arm $i = 1, \ldots, N$ set $k_i = 0, \hat{\mu}_i = 0$
2: **for** $t = 1, 2, \ldots,$ **do**
3:     For each arm $i = 1, \ldots, N$, sample $\theta_i(t)$ from the $\mathcal{N}\left(\hat{\mu}_i, \frac{1}{k_i+1}\right)$ distribution.
4:     Play arm $i(t) := \arg\max_i\{\theta_i(t)\}$ and observe reward $r_t$
5:     Set $\hat{\mu}_{i(t)} := \frac{\hat{\mu}_{i(t)}k_{i(t)}+r_t}{k_{i(t)}+1}, k_{i(t)} := k_{i(t)} + 1$
6: **end for**
---

---
**Algorithm 4** Thompson Sampling for Linear Contextual Bandits
---
1: Set $B(1) = I_d, \hat{\mu} = 0_d, f = 0_d$.
2: **for** $t = 1, 2, \ldots,$ **do**
3:     Sample $\tilde{\mu}(t)$ from distribution $\mathcal{N}\left(\hat{\mu}, B(t)^{-1}\right)$.
4:     Play arm $i(t) := \arg\max_i x_i(t)^T \tilde{\mu}(t)$, and observe reward $r_t$.
5:     Update $B(t+1) = B(t) + x_i(t)x_i(t)^T, f = f + x_i(t)r_t, \hat{\mu} = B(t)^{-1}f$.
6: **end for**
---

---
**Algorithm 5** Robust UCB
---
1: **Params**: Robustness parameter $\bar{C}$
2: **Init**: Select each arm $i = 1, \ldots, N$ for once, observe reward $r_i$, set $k_i = 1, \hat{\mu}_i = r_i$
3: **for** $t = N + 1, \ldots, T,$ **do**
4:     Play arm $i(t) = \arg\max_i \hat{\mu}_i + \sqrt{\frac{2\log t}{k_{i(t)}}} + \frac{\bar{C}}{k_{i(t)}}$
5:     Set $\hat{\mu}_{i(t)} := \frac{\hat{\mu}_{i(t)}k_{i(t)}+r_t}{k_{i(t)}+1}, k_{i(t)} := k_{i(t)} + 1$
6: **end for**
---

