# OpenReview forum: "Robust Thompson Sampling Algorithms Against Reward Poisoning Attacks"
_ICLR.cc/2025/Conference — Submitted to ICLR 2025_

### Official Review · Reviewer_CZzm · 2024-11-01

**Soundness:** 2
**Presentation:** 1
**Contribution:** 2
**Rating:** 3
**Confidence:** 5

**Summary:**

This paper studies the performance of Thompson Sampling for stochastic Gaussian bandits and contextual Gaussian $d$-dimensional linear bandit problems with Gaussian prior. Starting from the prior distribution over the "reward-parameter", the Thompson Sampling algorithm works by randomly selecting actions according to their posterior probability of being optimal. More specifically, at each time step, it samples a "reward-parameter" estimate from the posterior distribution conditioned on the history and selects the optimal action for the sampled parameter estimate given the context.

In this work, the authors study the case where the rewards can be corrupted by an adversary with full knowledge of the "reward-parameter", the history, the current context and the random reward before attack. The adversary perturbs the reward with an additive perturbation. The sum of absolute perturbation, $C$, is referred to as the budget of the attack or the corruption level and is assumed to be finite. The authors propose two modifications of the Thompson Sampling to mitigate the effects of the adversarial attacks on the algorithm's performance. The first algorithm, developed for Gaussian stochastic bandits with Gaussian prior, uses an optimistic biased distribution over the reward-parameters to compensate for potential reward attacks. The authors claim that they have derived a bound of order $O(\sqrt{NT \log(N)} + N\overline{C} + N)$ on the expected regret, where $N$ is the number of arms, $T$ is the number of time steps and $\overline{C}$ is the "robustness level", a parameter of the algorithm that is assumed to be greater than $C$. It has to be noted that the constant term in the bounds is of the order $10^{29}$.

The second algorithm, developed for contextual Gaussian $d$-dimensional linear bandit problems with Gaussian prior,  works by weighting the posteriors updates inversely proportional to the weighted 2-norm of the current context.  The authors derive a probabilistic bound on the expected regret.


The authors then perform experiments to demonstrate the performance of their modified Thompson Sampling algorithms. The first experiment compares the performance of their first algorithm against UCB and TS for Gaussian stochastic bandits with Gaussian priors. The second experiment compares the performance of their first algorithm against UCB and TS for contextual Gaussian $d$-dimensional linear bandit problems with Gaussian prior.

**Strengths:**

The main strength of the paper is to propose modifications to the Thompson Sampling algorithm to mitigate the effects of adversarial reward attacks.

**Weaknesses:**

Although adapting the Thompson Sampling to adversarial reward attacks is an interesting idea, the paper suffers from several weaknesses listed below (the order does not reflect a hierarchy in the severity of the weaknesses).

 - The first weakness concerns the considered settings. The authors considered a very limited settings as they assume that "the priors for the rewards of the arms and the reward parameters are Gaussian distributions" [lines 172-173]. They later claim that their algorithm can be "in principle" extended to general priors without giving any detail or example to support their claim. More problematic, their derived regret bounds (Theorem 4.1 and 5.1) rely extensively on those assumptions and do not hold for general priors. This limitation regarding the scope of the paper needs to be included in the title, the abstract, and the introduction.
 - A second weakness concerns the result derived in Theorem 4.1. Following the last part of the proof in the Appendix, the authors forgot to sum over all the possible $ N $ arms. Taking the sum over all possible arms leads to a regret bound of order $O(N^{3/2}\sqrt{T \log(N)} + N^{2}\overline{C} + N^{2})$. Another concern regarding the regret bound is the enormous multiplicative constant $72(e^{64}+5) \approx 10^{29}$, which raises the question of whether this bound can be applicable in practice.
- Another weakness concerns the clarity of the paper and the presentation of the proofs. Notations are at best cumbersome (e.g., $x_{i(t)}(t)^T\mu$), sometimes incoherent (e.g., the history denoted $H_{t-1}$ in the paper and later changed to $F_{t-1}$ in the appendix, mixing notations $\theta$ and $\Theta$), incomplete (e.g., summation symbols without above limit $\sum_{t=1}$ [line 133]), or simply not defined (e.g., $\tau_k$, $\Delta_i$, $\overline{E_i^\theta(t)}$, $\overline{E_i^\mu}(t)$, $c_s$, $L_i(T)$). This seriously hinders the clarity of the paper and the ability to evaluate the proof's correctness. Although the single column allows enough space to write on a single line, authors often split equations or even expressions. Examples can be found on lines 311 to 315 or 684 to 690. The authors also use overfull lines, such as on line 1033. Often, proofs consist simply of a long chain of equalities and inequalities with no justification given to the reader. Sometimes, the authors cite results from previous papers without providing any reference.  Overall, the appendix seems to have been written to discourage readers or make it impossible for them to verify the proof.
- One more weakness concerns the writing of the paper. This reviewer believes that at least parts of the paper have been written by generative AI tools. Indeed, parts of the paper present typical AI writing patterns, such as repetition of ideas and phrases, overuse of specific writing structures, and generic or high-level descriptions. One example is in the introduction between lines 36 and 47:
    "[...]Thompson sampling has several advantages:
  - Utilizing prior information: By design, Thompson sampling algorithms utilize and benefit from the prior information about the arms.
  - Easy to implement: While the regret of a UCB algorithm depends critically on the specific choice of upper-confidence bound, Thompson sampling depends only on the best possible choice. This becomes an advantage when there are complicated dependencies among actions, as designing and computing with appropriate upper confidence bounds present significant challenges Russo et al. (2018). In practice, Thompson sampling is usually easier to implement Chapelle \& Li (2011).
  - Stochastic exploration: Thompson sampling is a random exploration strategy, which could be more resilient under some bandit settings Lancewicki et al. (2021).
    Despite the success, Thompson sampling faces the problem of [...]"

- Another weakness concerns the experiments performed by the authors. First, one can deplore that the adversarial attacks are not explained. Second, for the stochastic bandit experiment, one can regret that the proposed algorithms are not compared against any other robust algorithm from the literature. Third, the plotted results raise serious questions regarding the quality of the experiments:
  - on Figure 1, the left and right plots seem to present the same data although their respective titles claim otherwise.
  - on Figure 1 and Figure 2, the expected regret of TS seems not just close but almost exactly equal to the expected regret of UCB.
  - on the top part of Figure 2, the titles of the $x$-axes are almost not readable.
  - on the bottom part of Figure 2, it seems impossible to explain how the expected regret of the proposed algorithm can decrease when the adversarial budget increases. This strongly suggests that there is a mistake in the experiments.
  - another incoherence in the displayed results that indicated an error in the experiments is the fact that on the top part of Figure 2, the robust version of Thompson Sampling seems to outperform the original Thompson Sampling and UCB algorithm \emph{even with no adversarial attack} (when the adversarial budget is $0$). This simple \emph{sanity check} strongly suggests some mistakes in the code.
  - there seem to be confidence intervals associated to each point on the plots but no information is given about the level of confidence they represent.
  - the code for running the experiments is provided without a readme.txt file, or a single comment explaining how to run the code or what is the purpose of the different files and functions. It was therefore not possible to verify the soundness of the experiments.

**Questions:**

Here is a list of suggestions for the authors.

- The first suggestion would be to entirely rewrite the proofs of Theorem 4.1 and Theorem 5.1 in the Appendix. Make sure to properly introduce all the notation with clear definitions to justify all the steps in your proofs (equalities, inequalities, the use of some lemma,….) to provide reference to all the lemmata you borrow from previous work. To the best of your capacity, try to write the proofs in a pedagogical way such that a reader familiar with the work can easily follow and asses the steps in the reasoning.
- The second suggestion is for the authors to verify and clarify the omission in the last step of Theorem 4.1’s proof regarding the summation over all arms.
\item The third suggestion is for the authors to provide more details supporting their claim that the algorithm and regret bounds can be extended to general priors. If this is not possible, the authors would have to change the title, abstract, and introduction to acknowledge the limited scope of the work.
\item A fourth suggestion concerns the experimental results. For the Gaussian stochastic bandit setting, the authors should compare the performance of the first suggested algorithm against a competitive, robust algorithm. Also, the authors should provide a detailed description of the adversarial attacks used in the experiments. Also, the authors should address the following issues in Figures 1 and 2:
   - are the data in the left and right plots of Figure 1 indeed distinct?
   - why the expected regret of TS is nearly identical to UCB?
   - clarification on the expected regret’s decrease as the adversarial budget increases on Figure 3.
   - the authors should explain the level of confidence for the plotted intervals.
   - finally, the authors should provide commented code with a description of the different files and functions and a readme.txt file so that a reviewer can reproduce the experiments quickly.
- A fifth suggestion would be for the authors to avoid using generative tools to write articles. It is frustrating for a reviewer to question whether they have spent more time writing a review than the author to write their paper.

---

> ### Author Response · Authors · 2024-11-23
>
> Thank you for your constructive comments and suggestions. Based on the reviews, the revision has been uploaded, and the most important changes are highlighted in blue. Below is our response to your concerns.
>
> Q1: The authors considered a very limited settings.
>
> A1: We have made it clear in the title, abstract, and contribution that we only consider Gaussian bandits. It is common in Thompson sampling studies to focus on a representation case of priors such as Gaussian distributions. [1,2,3].
>
> Q2: The authors forgot to sum over all the possible N Arms
>
> A2: In the proof of Theorem 4.1, we summed the regret upper bound over all possible N arms. We made a slight simplification in the summation notation($\sum_{i}$), and we will use the complete form in the revised version($\sum_{i=1}^{N}$). Specifically, as presented in the proof of Theorem 4.1:
> \begin{aligned}
>     &\mathbb{E}[\mathcal{R}(T)] = \sum_{i} \Delta_i \mathbb{E}\left[k_i(T)\right] \leq \sum_{i}[72\left(e^{64}+5\right) \frac {\ln \left(T \Delta^2_i\right)}{\Delta_i} + 12C+\frac{13}{ \Delta_i}+\Delta_i]
> \end{aligned}
>
> The equality on the left-hand side holds because $\mathbb{E}\left[k_i(T)\right]$ represents the expected number of times arm $i$ is pulled, and $\Delta_i$ is the regret incurred each time arm $i$ is pulled. Therefore, $\Delta_i \mathbb{E}\left[k_i(T)\right]$ represents the expected regret due to pulling arm $i$.
> The inequality on the right-hand side follows by combining the bounds from Lemmas A.3 to A.5, where we have:
> $$
> \begin{aligned}
> &\mathbb{E}\left[k_i(T)\right] \leq 72\left(e^{64}+4\right)\frac{\ln \left(T \Delta_i^2\right)}{\Delta_i^2}+\max [ \frac{12C}{\Delta_i},  \frac{32 \ln \left(T \Delta_i^2\right)}{\Delta_i^2} ]+\frac{13}{\Delta_i^2}+1
> \end{aligned}
> $$
> By multiplying by $\Delta_i$ and summing over all $N$ arms, we then obtain the desired inequality on the right-hand side.
>
> Q3: Another concern regarding the regret bound is the enormous multiplicative constant
>
> A3: We have noticed the problem. This is a motivation for our simulations. We empirically show that, in practice, the constant dependency of the regret is small. Add improvement
>
> Q4: the expected regret of TS seems not just close but almost exactly equal to the expected regret of UCB; it seems impossible to explain how the expected regret of the proposed algorithm can decrease when the adversarial budget increases
>
> A4: We have explained the two observations in detail in the revision:
>
> ‘We notice that the regret of the two vulnerable algorithms under the attack are almost identical. The reason is that the attacker highlights a target arm. For the vulnerable algorithms, they will believe that the target arm is optimal and almost always take that arm during training. So their behavior and regret are very similar under the attack.’
>
> ‘We notice that when the corruption level is small, the regret of our algorithm decreases slightly as the corruption level increases. The reason is that the attacker tries to highlight a randomly chosen target arm, and the arm is not the one with the lowest reward. Therefore, when the corruption level is not high enough to mislead the agent, the agent will only take sub-optimal arms for a limited number of rounds, and it tends to take the target arm more often instead of the worse arms, making the total regret slightly lower.’
>
> Q5: the titles of the x-axes are almost not readable
>
> A5: We have fixed the plots in the revision.
>
> Q6: The robust version of Thompson Sampling seems to outperform the original Thompson Sampling and UCB algorithm when the adversarial budget is 0.
>
> A6: The robust and original Thompson sampling are identical when the adversarial budget is 0. In the plots, their regrets are also the same when the adversarial budget is 0. It is also not a surprise that in our setup, the Thompson sampling algorithm performs slightly better than the UCB algorithm when there is no attack.
>
> Other Questions and suggestions about the writing:
>
> A: We have improved the proof in the appendix and made them more clear in the revision. We clarify in the revision that the bars in the plots are the variance of the data. We add a readme file to the supplementary materials. We clarify that no generative AI tools have been used in our writing. We run an AI scan on the text mentioned in the review using GPT Zero, and it says that with 98% accuracy, the text is fully written by humans. We have improved our writing in the revision.
>
> [1]: Honda, Junya, and Akimichi Takemura. "Optimality of Thompson sampling for Gaussian bandits depends on priors." Artificial Intelligence and Statistics. PMLR, 2014.
>
> [2]: Agrawal, Shipra, and Navin Goyal. "Thompson sampling for contextual bandits with linear payoffs." International conference on machine learning. PMLR, 2013.
>
> [3]:Hu, Bingshan, and Nidhi Hegde. "Near-optimal thompson sampling-based algorithms for differentially private stochastic bandits." Uncertainty in Artificial Intelligence. PMLR, 2022

---

> ### Author Response · Authors · 2024-11-26
>
> Dear Reviewer CZzm,
>
> Thanks again for your constructive reviews! We have revised the paper according to your questions and suggestions. Please let us know if you have any follow-up comments or concerns about our responses and revision. We are happy to answer any further questions.

---

> > ### Comment · Reviewer_CZzm · 2024-12-01
> >
> > Thank you for your response and for addressing some of my comments. The overall presentation of the paper has slightly improved. However, the revised version of the paper now exceeds the 12-page limit, which does not comply with the conference’s paper length requirements. It is unclear if this version should be considered for review.
> >
> > More importantly, the main weakness of the paper—namely, the enormous multiplicative constant in the regret analysis, despite the narrow setting of Gaussian linear bandits with Gaussian priors—remains. Additionally, the clarification provided by the authors regarding the expected regret’s decrease as the adversarial budget increases in Figure 3 is not satisfying.
> >
> > For all the above reasons, I maintain my score unchanged.

---

### Official Review · Reviewer_MHci · 2024-11-03

**Soundness:** 3
**Presentation:** 3
**Contribution:** 3
**Rating:** 6
**Confidence:** 3

**Summary:**

This paper considers poisoning attacks in multi-armed bandit setting. More specifically, the authors consider Thompson sampling algorithms and aim to design corruption-robust versions of these algorithms. They propose two algorithms, for stochastic and contextual linear settings, respectively, robust to poisoning attacks against an attacker that has limited corruption budget. The authors argue that their results are near-optimal, and they provide simulation results comparing their approaches to validate their theoretical findings.

**Strengths:**

- The paper is interesting and enjoyable to read. It clearly explains the most important ideas and positions its contributions well with respect to prior work on corruption-robust bandits.
- To my knowledge, prior work has not studied Thompson sampling algorithms that are robust to reward poisoning attacks. The proof techniques in the paper appear to be standard, but I found the analysis non-trivial. While I didn't check all the proofs in great detail, the arguments provided in the proof sketches appear sound.
- The proposed algorithms are simple extensions of existing methods and are based on adjusting the posteriors to make them less susceptible to manipulation. The upper bounds on the regret of the proposed algorithms are near-optimal, as argued in the paper, with the arguments relying on the lower bounds from prior work.
- The paper supports its theoretical results with experiments.

**Weaknesses:**

- While I found the results interesting, they are also somewhat expected, given the findings from prior work on poisoning attacks and corruption robustness in bandits.
- The experiments related to the stochastic MAB setting do not include a corruption-robust baseline. For the contextual bandit setting, the performance of the proposed method is similar to the corruption-robust baseline but is often worse. It would also be useful to include a richer set of attack strategies in the experiments.
- While I believe that the paper is overall well-written, the presentation could be improved and polished. There are quite a few typos in the text, and some of the figures in the experimental section do not adequately label the axes on the plots (see Figs. 2 and 4). Moreover, the fonts in the figures are too small. It also seems that some references are cited incorrectly. For example, the paper frequently cites He et al. (2023), which appears to be about *Nearly Minimax Optimal Reinforcement Learning for Linear Markov Decision Processes*. It would be helpful if the authors clarified this.

**Questions:**

I did not fully follow the intuition behind Algorithm 2 and the corresponding proof sketch. More specifically, it would be helpful if the authors elaborated further on their two remarks:
- *...the attacker can apply less influence on the estimator by corrupting such data...*,
- *...variance of the posterior distribution is limited due to the weighted estimator under reward poisoning...*

Please see *Strengths* and *Weaknesses* for my other comments and questions.

---

> ### Author Response · Authors · 2024-11-23
>
> Thank you for your constructive comments and suggestions. Based on the reviews, the revision has been uploaded, and the most important changes are highlighted in blue. Below is our response to your concerns.
>
> Q1: While I found the results interesting, they are also somewhat expected, given the findings from prior work on poisoning attacks and corruption robustness in bandits
>
> A1: One can expect that making Thompson sampling robust is feasible using the techniques in our work, but it is hard to know exactly how robust a Thompson sampling algorithm can be and if it is possible to achieve near-optimal guarantees in the cases where the corruption level $C$ is known or unknown to the agent. Our work shows how to make near-optimal robust Thompson sampling algorithms and proves that they can be near-optimal.
>
> Q2: The experiments related to the stochastic MAB setting do not include a corruption-robust baseline.
>
> A2: We have included a robust baseline for the stochastic MAB setting in the revision. The baseline we call `Robust UCB’ is essentially the standard UCB algorithm with an additional bonus term $C/k_i(t)$ to arm $i$ to compensate for the potential influence of data corruption, where $C$ is the corruption level, and $k_i(t)$ is the number of times arm $i$ being selected at time $t$. Such an idea is also mentioned in [1], and the details can be found in the revision in section 6.1. Our empirical results show that the performance of our robust Thompson sampling is similar to that of the robust UCB algorithm. Thompson sampling has several advantages over the UCB algorithm, as mentioned in our introduction section. Our empirical results further show that the robust Thompson sampling algorithms are competitive against robust UCB/OFUL algorithms.
>
> Q3: While I believe that the paper is overall well-written, the presentation could be improved and polished.
>
> A3: We have fixed the typos and plots in the revision.
>
> Q4: it would be helpful if the authors elaborated further on their two remarks:
> … the attacker can apply less influence on the estimator by corrupting such data...,
> ...variance of the posterior distribution is limited due to the weighted estimator under reward poisoning..
>
> A4: We have changed the two remarks in the revision:
>
> 1. ‘The key is that it assigns less weight $w_t$ to the data with a `large' context $w_{t}=\min (\{1, \gamma /\left\||x_{i(t)}(t)\right\||_{B(t)^{-1}}\})$
> so that its estimation is less sensitive to data corruption in these cases.’
>
> 2. ‘The key reason is that the posterior distribution computed by the weighted estimator is less sensitive to any change in the rewards. So the agent is more robust against data corruption.’
>
> [1]: Lykouris, Thodoris, Vahab Mirrokni, and Renato Paes Leme. "Stochastic bandits robust to adversarial corruptions." Proceedings of the 50th Annual ACM SIGACT Symposium on Theory of Computing. 2018.

---

> > ### Comment · Reviewer_MHci · 2024-11-26
> >
> > Thank you for the clarifications. I'm also happy to see that you've improved the revised version of the paper. Hence, I will keep my positive score.

---

### Official Review · Reviewer_nYyd · 2024-11-04

**Soundness:** 4
**Presentation:** 3
**Contribution:** 3
**Rating:** 6
**Confidence:** 3

**Summary:**

This paper proposes first near-optimal variants of Thompson sampling for stochastic bandit and contextual linear bandit problems with adversarial reward poisoning.

**Strengths:**

This is the first work providing variants of Thompson sampling for this class of problems.
Proposed algorithms are near-optimal and, being based on Thompson sampling, they inherit the advantages of Thompson sampling over approaches based on optimism in face of uncertainty.

**Weaknesses:**

Previous works are mentioned in the introduction and related works section. However, comparing existing results requires finding each cited paper and going through them one by one. It would be better if a table was included with previous works.

typos:
At the end of Sections 4 and 5, the paper (He et al., 2022) should be cited instead of (He et al., 2023).

**Questions:**

What can be said about Bayesian regret?

---

> ### Author Response · Authors · 2024-11-23
>
> Thank you for your constructive comments and suggestions. Based on the reviews, the revision has been uploaded, and the most important changes are highlighted in blue. Below is our response to your concerns.
>
> Q1: However, comparing existing results requires finding each cited paper and going through them one by one. It would be better if a table was included with previous works
>
> A1: We have included a table in the revision to show the comparison of the theoretical results between our algorithms and current state-of-the-art algorithms.
>
> Q2: typos: At the end of Sections 4 and 5, the paper (He et al., 2022) should be cited instead of (He et al., 2023)
>
> A2: We have fixed them in the revision.
>
> Q3: What can be said about Bayesian regret?
>
> A3: In the stochastic setting, we have demonstrated that for any parameter vector
> $\mu = (\mu_1, \cdots, \mu_N) \in \Pi_{i=1}^{N} [0,1]$
> the expected regret is bounded by
> $ \sum_{i=1}^{N} \left[ 72 \left(e^{64} + 5 \right) \frac{\ln \left( T \Delta_i^2 \right)}{\Delta_i} + 12C + \frac{13}{\Delta_i} + \Delta_i \right], $
> where $\Delta_i$ represents the sub-optimality gap for arm $i$. Therefore, the analysis can be divided into two cases: $\Delta_i \geq e \sqrt{\frac{N \ln N}{T}}$ and $\Delta_i < e \sqrt{\frac{N \ln N}{T}}$. There exists a constant $k$(e.g. $k=144 \left(e^{64} + 5 \right) $) independent of $\mu$ such that
> $ \mathbb{E}[\mathcal{R}(T) \mid \mu] \leq k(\sqrt{N T \ln N} + N C + N) $
> holds for any $\mu \in \Pi_{i=1}^{N} [0,1]$. Consequently, the Bayesian regret satisfies
> $ \mathbb{E}_{\mu}[\mathbb{E}[\mathcal{R}(T) \mid \mu]] \leq k(\sqrt{N T \ln N} + N C + N). $
>
> This result establishes a uniform bound on the expected regret across all possible values of $\mu$, ensuring the validity of our regret analysis in the Bayesian context.

---

> > ### Comment · Reviewer_nYyd · 2024-11-26
> >
> > Thank you for the response. I think including a discussion of Bayesian regret will improve the article. I will keep my positive score.

---

### Official Review · Reviewer_skgH · 2024-11-06

**Soundness:** 2
**Presentation:** 3
**Contribution:** 2
**Rating:** 3
**Confidence:** 5

**Summary:**

This work studies the setting where the revealed rewards are corrupted. So, the learning agent cannot use the true reward to do posterior sampling. They consider two bandit variants with corruption: stochastic bandit and linear contextual bandit. They also propose efficient algorithms.

**Strengths:**

a new problem

**Weaknesses:**

problem set up: I still think the proposed problem is a special case (actually a simpler case) of differentially private online learning, based on lines 149 to 151.

proposed algorithms: they is not that interesting nor novel, simply re-shaping the posterior distribution in an optimistic way. This idea has been used in Hu and Hedge, 2022.

**Questions:**

1. what does not mean of "optimality in the face of corruption" in line 65?

2. In line 795, the constant is 72 e^{64}. You can improve the constant by using some analysis in https://arxiv.org/abs/2407.14879.

Also, I think the aforementioned paper is quite related to this work.

---

> ### Author Response · Authors · 2024-11-23
>
> Thanks for your constructive comments and suggestions. Based on the reviews, the revision has been uploaded, and the most important changes are highlighted in blue. Below is our response to your concerns.
>
> Q1: I still think the proposed problem is a special case (actually a simpler case) of differentially private online learning, based on lines 149 to 151; This idea has been used in [1]
>
> A1: Differentially private setting can be thought of as a robustness against poisoning attack setting. The attacker can modify a certain number of data points, and a differentially private algorithm needs to ensure that its behavior is similar under any possible attacks. However, there are also two main differences between the two settings.
>
> 1. Robustness goal: in the differentially private setting, the goal of the agent is to ensure that the decisions it makes under corruption are similar to the decisions without corruption. As defined in [1], let $M$ be the algorithm and $X, X’$ be two neighboring reward sequences to represent the original and corrupted rewards. Then the agent needs to ensure that $P\{M(X) \in D\} \leq e^\epsilon P\{M(X’) \in D\}$ holds for any decision set $D$ and a small value of $\epsilon$; In the reward poisoning attack setting considered in our work, the goal of the learning agent is to minimize the total regret/ maximize the total revenue $\sum_{t} r(t)$.
>
> 2. Attack constraint: Let $c(t)$ be the corruption at time $t$. In the differentially private setting, the attacker is constrained by the total number of data being corrupted $\sum \mathbb{1}[c(t) \neq 0] \leq C$; In the reward poisoning attack setting considered in our work, the attacker is constrained by the total amount of corruption $\sum_{t} |c(t)| \leq C$. Note that there is no constraint on the number of data being corrupted in the reward poisoning setting, and it can be as large as the number of all data: $\sum \mathbb{1}[c(t) \neq 0]$=T$.
>
> Therefore, the poisoning attack setting is not a special case in a differentially private setting.
>
> As a result, a differentially private algorithm like the one in [1] is not necessarily a robust algorithm against reward poisoning attacks. A differentially private algorithm can only guarantee that when some of the data are corrupted, the regret of the algorithm remains low. In the reward poisoning attack setting, even under a limited corruption budget $\sum_{t} |c(t)| \leq C$, every single data point can be corrupted: $\sum \mathbb{1}[c(t) \neq 0]=T$. For an $\epsilon$-differentially private algorithm proposed in [1], for two reward sequences $X$ and $X’$ that differ everywhere, it can only guarantee that $P\{M(X) \in D\} \leq e^{\epsilon \cdot T} P\{M(X’) \in D\}$. Setting $\epsilon=o(1/T)$ can make the guarantee non-trivial, but it also results in $\Omega(T)$ regret for the algorithm. It has also been theoretically proved in a different learning setting that the differentially private algorithms are vulnerable to data poisoning attacks because every single data point can be modified under the attack [2].
>
>
> Q2: what does "optimality in the face of corruption" mean in line 65?
>
> A2: "optimality in the face of corruption" is a general idea similar to the popular exploration strategy “optimality in the face of uncertainty.” In “optimality in the face of uncertainty,” the agent optimistically estimates the best possible reward for an arm considering the uncertainty in its evaluation. Similarly, in "optimality in the face of corruption", the agent optimistically estimates the best possible reward for an arm considering the uncertainty and the influence of data corruption on its estimation.
>
> Q3: In line 795, the constant is 72 e^{64}. You can improve the constant by using some analysis in https://arxiv.org/abs/2407.14879.
>
> A3: Thanks for the suggestion. We find that it is not straightforward to directly apply the techniques in this work to improve the big constant in our bound, but we will follow the idea in that work and improve our theoretical results in future revisions.
>
> [1]: Hu, Bingshan, and Nidhi Hegde. "Near-optimal thompson sampling-based algorithms for differentially private stochastic bandits." Uncertainty in Artificial Intelligence. PMLR, 2022
>
> [2]: Ma, Yuzhe, Xiaojin Zhu, and Justin Hsu. "Data poisoning against differentially-private learners: Attacks and defenses." arXiv preprint arXiv:1903.09860 (2019).

---

> ### Author Response · Authors · 2024-11-26
>
> Dear Reviewer skgH,
>
> Thanks again for your constructive reviews! We have revised the paper according to your questions and suggestions. Please let us know if you have any follow-up comments or concerns about our responses and revision. We are happy to answer any further questions.

---

> ### Author Response · Authors · 2024-11-30
>
> Thanks again for your valuable comments. As the discussion period is ending, we'd love to know if our response addresses your concerns.
>
> In addition to our previous response, we find that the work mentioned in your review [1] has already proved that the Thompsom sampling algorithm for MAB is differentially private as-is. At the same time, it is not adversarially robust, as mentioned in our work. This clearly demonstrates that adversarial robustness is not a special case of differentially private online learning, and a differentially private algorithm may not be robust against adversarial data poisoning attacks.
>
> [1]: Ou, Tingting, Rachel Cummings, and Marco Avella. "Thompson Sampling Itself is Differentially Private." International Conference on Artificial Intelligence and Statistics. PMLR, 2024.

---

### Meta-Review · Area_Chair_LboR · 2024-12-21

**Metareview:**

This paper examines modified Thompson Sampling algorithms for Gaussian bandits and contextual linear bandits with Gaussian priors, focusing on mitigating adversarial attacks that perturb rewards. The first modification introduces an optimistic biased distribution for Gaussian stochastic bandits, while the second adjusts posterior updates for contextual linear bandits based on the current context. Both methods aim to reduce expected regret under adversarial conditions. Experiments show the effectiveness of these modifications compared to UCB and Thompson Sampling.

While the effort put by the authors to improve the paper is commendable, as pointed out by the reviewers, it's hard to judge the correctness of the proofs and efficacy of the results with these many modifications. Moreover, there are remaining concerns about the correctness of the proofs and the nature of the empirical results (e.g., Fig 3), which are non-intuitive. Based on these, I suggest that the authors submit to the next suitable venue after incorporating all the reviewer feedback.

**Additional Comments On Reviewer Discussion:**

See above.

---

### Decision · Program_Chairs · 2025-01-22

Reject